# SINKHORN DISCREPANCY FOR COUNTERFACTUAL GENERALIZATION

## ABSTRACT

Estimating individual treatment effects from observational data is highly challenging due to the existence of treatment selection bias. Most prevalent approaches mitigate this issue by aligning distributions of different treatment groups in the representation space. However, there are two critical problems circumvented: (1) mini-batch sampling effects (MSE), where the alignment easily fails due to the outcome imbalance or outliers at a mini-batch level; (2) unobserved confounder effects (UCE), where the unobserved confounders damage the correct alignment. To tackle these problems, we propose a principled approach named **E**ntire **S**pace **C**ounter**F**actual **R**egression (ESCFR) based on a generalized sinkhorn discrepancy for distribution alignment within the stochastic optimal transport framework. Based on the framework, we propose a relaxed mass-preserving regularizer to address the MSE issue and design a proximal factual outcome regularizer to handle the UCE issue. Extensive experiments demonstrate that our proposed ESCFR can successfully tackle the treatment selection bias and achieve significantly better performance than state-of-the-art methods.

## 1 INTRODUCTION

Estimating individual treatment effect (ITE) with randomized controlled trials is a common practice in causal inference, which has been widely used in e-commerce (Betlei et al., 2021), education (Cordero et al., 2018), and health care (Schwab et al., 2020). For example, drug developers would conduct clinical A/B tests to evaluate the drug effects. Although randomized controlled trials are the gold standard (Pearl & Mackenzie, 2018) for causal inference, it is often prohibitively expensive to conduct such experiments. Hence, observational data that can be acquired without intervention has been a tempting shortcut. For example, drug developers tend to assess drug effects with post-marketing monitoring reports instead of clinical A/B trials. With the growing access to observational data, estimating ITE from observational data has attracted intense research interest.

Estimating ITE with observational data has two main challenges: (1) missing counterfactuals, *i.e.*, only one factual outcome out of all potential outcomes can be observed; (2) treatment selection bias, *i.e.*, individuals have their preferences for treatment selection, making units in different treatment groups heterogeneous. To handle missing counterfactuals, meta-learners (Künzel et al., 2019) decompose the ITE estimation task into solvable factual outcome estimation subproblems. However, the treatment selection bias makes it difficult to generalize the factual outcome estimators trained within respective treatment groups to the entire population; consequently, the derived ITE estimator is biased.

Beginning with counterfactual regression (Shalit et al., 2017) and its revolutionary performance, most prevalent methods handle the selection bias by minimizing the distribution discrepancy between groups in the representation space (see Liuyi et al., 2018; Hassanpour & Greiner, 2020; Cheng et al., 2022). However, two critical issues with these methods have long been neglected, which significantly impedes them from handling the treatment selection bias. The first problem is the mini-batch sampling effects (MSE). Specifically, current representation-based methods (Shalit et al., 2017; Liuyi et al., 2018) compute distribution discrepancy within mini-batches instead of the entire data space, making it vulnerable to bad sampling cases. For example, given two aligned distributions, if a mini-batch outlier exists in the sampled distribution, the mini-batch discrepancy will be significant, making the training process noise-filled. The second problem is the unobserved confounder effects

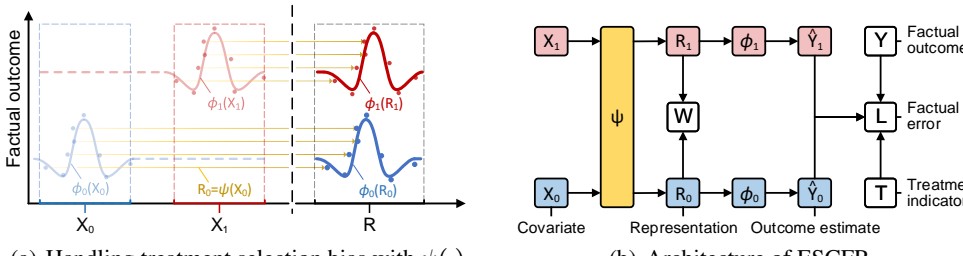

(a) Handling treatment selection bias with $\psi(\cdot)$.

(b) Architecture of ESCFR.

Figure 1: Overview of handling treatment selection bias with ESCFR. Red (blue) indicates treated (untreated) group. (a) Treatment selection bias causes the shift between $X_1$ and $X_0$, impeding $\phi_1$ and $\phi_0$ to generalize beyond the respective group's properties. Scatters and curves indicate the units and fitted outcome mappings, respectively. (b) ESCFR handles this issue by mapping covariates to an overlapped representation space with $R = \psi(X)$ where $\phi_1$ and $\phi_0$ are mutually generalizable.

(UCE). Specifically, current approaches directly assume *unconfoundedness* Ma et al. (2022), while the unobserved confounders widely exist in real scenarios and make the resulting estimators biased.

**Contributions and outline.** In this paper, we propose an effective ITE estimator based on optimal transport, Entire Space CounterFactual Regression (ESCFR), which tackles both the MSE and UCE issues with a generalized sinkhorn discrepancy. Specifically, after preliminaries in Section 2, we first reformulate the ITE estimation problem as a stochastic optimal transport problem in Section 3.1. We next showcase the MSE issue faced by existing approaches in Section 3.2 and propose a relaxed mass-preserving regularizer to mitigate this issue. We further investigate the UCE issue in Section 3.3 and propose a proximal factual outcome regularizer to solve it. We finally formulate the architecture and learning objectives of ESCFR in Section 3.4, and report the experimental results in Section 4.

## 2    PRELIMINARIES

### 2.1    CAUSAL INFERENCE FROM OBSERVATIONAL DATA

This section formulates basic definitions and models in observational causal inference. We first formalize the fundamental elements in Definition 2.1 following the general notation convention[1].

**Definition 2.1.** *Let $X$ be the random variable of covariates, with support $\mathcal{X}$ and distribution $\mathbb{P}(x)$; Let $R$ be the random variable of induced representations, with support $\mathcal{R}$ and distribution $\mathbb{P}(r)$; Let $Y$ be the random variable of outcomes, with support $\mathcal{Y}$ and distribution $\mathbb{P}(y)$; Let $T$ be the random variable of treatment indicator, with support $\mathcal{T} = \{0,1\}$ and distribution $\mathbb{P}(T)$.*

Following the potential outcome framework (Rubin, 1974), an individual with covariates $x$ has two potential outcomes, namely $Y_1(x)$ given it is treated and $Y_0(x)$ otherwise. The ground-truth individual treatment effect (ITE) is always formulated as the expected difference of potential outcomes:

$$\tau(x) := \mathbb{E}\left[Y_1 - Y_0 \mid x\right], \tag{1}$$

where one of these two outcomes is always unobserved. To address such missing counterfactuals, the ITE estimation task is commonly decomposed into potential outcome estimation subproblems that are solvable with any supervised learning method (Künzel et al., 2019). For example, T-learner models the factual outcomes $Y$ for units in the treated and untreated groups separately; S-learner regards the treatment indicator $T$ as one of the covariates $X$, and models $Y$ for all units simultaneously. The ITE estimate is then the difference of the estimated outcomes when $T$ is set to treated and untreated.

**Definition 2.2.** *Let $\psi : \mathcal{X} \to \mathcal{R}$ be a mapping from support $\mathcal{X}$ to $\mathcal{R}$, i.e., $\forall x \in \mathcal{X}, \exists r = \psi(x) \in \mathcal{R}$. Let $\phi : \mathcal{R} \times \mathcal{T} \to \mathcal{Y}$ be a mapping from support $\mathcal{R} \times \mathcal{T}$ to $\mathcal{Y}$, i.e., it maps the representations and treatment indicator to the corresponding factual outcome. For example, $Y_1 = \phi_1(R)$, $Y_0 = \phi_0(R)$, where we abbreviate $\phi(R, T = 1)$ and $\phi(R, T = 0)$ to $\phi_1(R)$ and $\phi_0(R)$, respectively, for brevity.*

---

[1]We use uppercase letters, *e.g.*, $X$ to denote a random variable, and lowercase letters, *e.g.*, $x$ to denote an associated specific value. Letters in calligraphic font, *e.g.*, $\mathcal{X}$ represent the support of the corresponding random variable, and $\mathbb{P}()$ represents the probability distribution of the random variable, *e.g.*, $\mathbb{P}(X)$.

TARNet (Shalit et al., 2017) obtains better performance by absorbing the advantages of both T-learner and S-learner, consisting of a representation mapping $\psi$ and an outcome mapping $\phi$ as defined in Definition 2.2. For an individual with covariates $X$, TARNet estimates ITE as:

$$\hat{\tau}_{\psi,\phi}(X) = \hat{Y}_1 - \hat{Y}_0, \quad \text{where} \quad \hat{Y}_1 = \phi_1(\psi(X)), \quad \hat{Y}_0 = \phi_0(\psi(X)), \tag{2}$$

where $\psi$ is trained over all individuals, $\phi_1$ and $\phi_0$ are trained over the treated and untreated group, respectively, to minimize the factual error. Finally, the performance of ITE estimators is mainly evaluated with the precision in estimation of heterogeneous effect (PEHE):

$$\epsilon_{\mathrm{PEHE}}(\psi,\phi) := \int_{\mathcal{X}} \left(\hat{\tau}_{\psi,\phi}(x) - \tau(x)\right)^2 \hat{\mathbb{P}}(x)\, dx. \tag{3}$$

However, according to Figure 1(a), the treatment selection bias causes a distribution shift of covariates across groups, which misleads $\phi_1$ and $\phi_0$ to overfit their respective group's properties and generalize poorly to the entire population. Therefore, the ITE estimate $\hat{\tau}$ by these methods would be biased.

## 2.2 DISCRETE OPTIMAL TRANSPORT AND SINKHORN DIVERGENCE

Optimal transport (OT) instantiates distribution discrepancy as the minimum transport cost, which provides a grip for quantifying the treatment selection bias in Figure 1(a). Monge (1781) first formulated OT as finding an optimal mapping between two distributions. However, this formulation cannot guarantee the existence and uniqueness of solutions. Kantorovich (2006) proposed a more applicable formulation in Definition 2.3, which can be seen as a generalization of Monge problem.

**Definition 2.3.** *For empirical distributions $\alpha$ and $\beta$ with n and m units, respectively, the Kantorovich problem aims to find a feasible plan $\pi \in \mathbb{R}_+^{n\times m}$ which transports $\alpha$ to $\beta$ at minimum cost:*

$$\mathbb{W}(\alpha,\beta) := \min_{\boldsymbol{\pi}\in\Pi(\alpha,\beta)} \langle \mathbf{D},\boldsymbol{\pi}\rangle, \quad \Pi(\alpha,\beta) := \left\{\boldsymbol{\pi}\in\mathbb{R}_+^{n\times m}: \boldsymbol{\pi}\mathbf{1}_m = \mathbf{a}, \boldsymbol{\pi}^{\mathrm{T}}\mathbf{1}_n = \mathbf{b}\right\}, \tag{4}$$

*where $\mathbb{W}(\alpha,\beta) \in \mathbb{R}$ is the Wasserstein discrepancy between $\alpha$ and $\beta$; $\mathbf{D} \in \mathbb{R}_+^{n\times m}$ is the unit-wise distance[2] between $\alpha$ and $\beta$; $\mathbf{a}$ and $\mathbf{b}$ indicate the mass of units in $\alpha$ and $\beta$, and $\Pi$ is the feasible transportation plan set which ensures the mass-preserving constraint holds.*

However, exact solutions (Bonneel et al., 2011) to (4) always come with high computational costs. As such, researchers would always add an entropic regularization to the Kantorovich problem:

$$\mathbb{W}^\epsilon(\alpha,\beta) := \langle \mathbf{D},\boldsymbol{\pi}^\epsilon\rangle, \quad \boldsymbol{\pi}^\epsilon := \arg\min_{\boldsymbol{\pi}\in\Pi(\alpha,\beta)} \langle \mathbf{D},\boldsymbol{\pi}\rangle - \epsilon \mathrm{H}(\boldsymbol{\pi}), \quad \mathrm{H}(\boldsymbol{\pi}) := -\sum_{i,j}\boldsymbol{\pi}_{i,j}\left(\log(\boldsymbol{\pi}_{i,j}) - 1\right), \tag{5}$$

making the problem $\epsilon$-convex and solvable with the Sinkhorn algorithm (Cuturi, 2013). The Sinkhorn algorithm only consists of matrix-vector products, making it suited to be accelerated with GPUs.

## 3 PROPOSED METHOD

In this section, we present the proposed Entire Space CounterFactual Regression (ESCFR) approach based on optimal transport to tackle the treatment selection bias. We first illustrate the stochastic optimal transport framework for distribution discrepancy minimization across treatment groups. Based on the framework, we then propose a relaxed mass-preserving regularizer to address the sampling effect, and a proximal factual outcome regularizer to handle the unobserved confounders. We finally summarize the model architecture, learning objectives, and optimization algorithm.

### 3.1 STOCHASTIC OPTIMAL TRANSPORT FOR COUNTERFACTUAL REGRESSION

Representation-based approaches mitigate the treatment selection bias by calculating the distribution discrepancy in the representation space and then minimizing it. We select optimal transport to compute the discrepancy since it has shown compelling advantages over its competitors. Specifically, it accounts for the distribution's geometry and thus works in the cases where $\phi$-divergence (*e.g.*, Kullback-Leibler divergence) fails (Seguy et al., 2018). In addition, the calculated discrepancy can

---

[2]We calculate the unit-wise distance with the squared Euclidean metric following Courty et al. (2017b).

be optimized with the traditional supervised learning framework instead of the adversarial learning framework, and is therefore easier to optimize than adversarial-based methods (Kallus, 2020).

Optimal transport calculates the group discrepancy as $\mathbb{W}\left(\mathbb{P}_\psi^{T=1}(r), \mathbb{P}_\psi^{T=0}(r)\right)$, where $\mathbb{P}_\psi^{T=1}(r)$ and $\mathbb{P}_\psi^{T=0}(r)$ are the distributions of representations in treated and untreated groups, respectively, induced by the mapping $r = \psi(x)$. The discrepancy is differentiable with respect to $\psi$ (Flamary et al., 2021), thus can be minimized by updating the representation mapping $\psi$ with gradient-based optimizers.

**Definition 3.1.** *Let $\hat{\mathbb{P}}^{T=1}(x)$ and $\hat{\mathbb{P}}^{T=0}(x)$ be the empirical distributions of covariates at a mini-batch level, which contain $n$ treated units and $m$ untreated units, respectively; $\hat{\mathbb{P}}_\psi^{T=1}(r)$ and $\hat{\mathbb{P}}_\psi^{T=0}(r)$ be that of representations induced by the representation mapping $r = \psi(x)$ defined in Definition 2.2.*

However, since prevalent neural estimators mainly update parameters with stochastic gradient methods, only a fraction of the units is accessible within each iteration. A shortcut in this context is to calculate the group discrepancy at a stochastic mini-batch level:

$$\hat{\mathbb{W}}_\psi := \mathbb{W}\left(\hat{\mathbb{P}}_\psi^{T=1}(r), \hat{\mathbb{P}}_\psi^{T=0}(r)\right). \tag{6}$$

To further investigate the effectiveness of this shortcut, Theorem 3.1 demonstrates that PEHE can be optimized by iteratively minimizing the factual outcome estimation error and the mini-batch group discrepancy (6). The proof of the theorem can be found in Appendix A.3.

**Theorem 3.1.** *Let $\psi$ and $\phi$ be the representation mapping and factual outcome mapping, respectively; $\hat{\mathbb{W}}_\psi$ be the group discrepancy at a mini-batch level. With the probability of at least $1 - \delta$, we have:*

$$\epsilon_{\mathrm{PEHE}}(\psi, \phi) \leq 2\left[\epsilon_{\mathrm{F}}^{T=1}(\psi, \phi) + \epsilon_{\mathrm{F}}^{T=0}(\psi, \phi) + B_\psi \hat{\mathbb{W}}_\psi - 2\sigma_Y^2 + \mathcal{O}\left(\frac{1}{\delta N}\right)\right], \tag{7}$$

*where $\epsilon_{\mathrm{F}}^{T=1}$ and $\epsilon_{\mathrm{F}}^{T=0}$ are the expected errors of factual outcome estimation, $N$ is the batch size, $\sigma_Y^2$ is the variance of outcomes, $B_\psi$ is a constant term, and $\mathcal{O}(\cdot)$ is a sampling complexity term.*

## 3.2 RELAXED MASS-PRESERVING REGULARIZER FOR SAMPLING EFFECT

Although Theorem 3.1 guarantees that the empirical OT discrepancy (6) bounds the PEHE, the sampling complexity term $\mathcal{O}(\cdot)$ inspires us to investigate the potential risks of bad cases caused by stochastic sampling. Precisely, $\mathcal{O}(\cdot)$ results from the discrepancy between the entire population and the sampled mini-batch units (see (30) and (32) in Appendix A) which is highly dependent on the uncontrollable sampling quality. Therefore, the discrepancy measure should be robust to bad sampling cases, otherwise the resulting huge variance will impede it from computing and reducing the actual discrepancy.

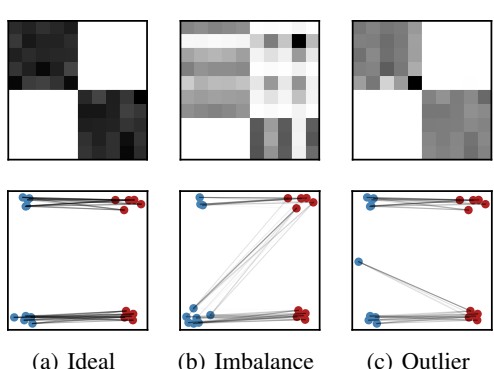

(a) Ideal     (b) Imbalance     (c) Outlier

Figure 2: Optimal transport plan (upper) and its geometric interpretation (down) in three cases, where the connection strength depicts the transported mass. Different colors (vertical positions) indicate different treatments (outcomes).

The OT discrepancy in (6) can be easily disturbed by many sampling cases, except for the ideal case in Figure 2(a) where the transport strategy is reasonable and applicable. For example, according to Figure 2(b), it falsely matches units with unrelated factual outcomes in the mini-batch where the outcomes between groups are imbalanced; according to Figure 2(c), it falsely matches the mini-batch outliers to normal units, causing a substantial disruption of the transportation strategy. In short, the vanilla OT technique in (6) fails to quantify the group discrepancy for producing erroneous transport strategies in non-ideal mini-batches, and thus misleads the update of the representation mapping $\psi$. We summarize this phenomenon as the mini-batch sampling effect (MSE) issue [3].

---

[3] Apart from OT, most prevalent methods (Ma et al., 2022; Liuyi et al., 2018) fail to handle the treatment selection bias for neglecting the MSE issue. The ability to formalize the MSE issue through the mass-preserving constraint is an important advantage of OT over other techniques, as it provides a grip for handling it.

This issue is attributed to the mass-preservation constraint in (5), which requires that all units in both groups match each other, regardless of the actual situation. Mini-batch outliers, for instance, would be compelled to be transported according to Figure 2, which impedes the transport of normal units and the computation of the actual group discrepancy. A small batch size would exacerbate this defect.

**Definition 3.2.** *For empirical distributions $\alpha$ and $\beta$ with n and m units, respectively, optimal transport with relaxed mass-preserving constraint seeks the transport strategy $\boldsymbol{\pi}$ at the minimum cost:*

$$\mathbb{W}^{\epsilon,\kappa}(\alpha,\beta) := \min_{\boldsymbol{\pi}} \langle \mathbf{D}, \boldsymbol{\pi} \rangle, \boldsymbol{\pi} := \arg\min_{\boldsymbol{\pi}} \langle \mathbf{D}, \boldsymbol{\pi} \rangle - \epsilon H(\boldsymbol{\pi}) + \kappa(\mathrm{D}_{\mathrm{KL}}(\boldsymbol{\pi}\mathbf{1}_m, \mathbf{a}) + \mathrm{D}_{\mathrm{KL}}(\boldsymbol{\pi}^{\mathrm{T}}\mathbf{1}_n, \mathbf{b})) \quad (8)$$

*where $\mathbf{D} \in \mathbb{R}_{+}^{n \times m}$ is the unit-wise distance, and $\mathbf{a}$; $\mathbf{b}$ indicate the mass of units in $\alpha$ and $\beta$, respectively.*

An intuitive approach to mitigate MSE is to relax the marginal constraint and allow for the creation and destruction of unit's mass. To this end, a relaxed mass-preserving regularizer (RMPR) is devised in Definition 3.2, which replaces the hard marginal constraint in (4) with a soft penalty in (8) for deriving transport strategy. In this context, the stochastic discrepancy is calculated as

$$\hat{\mathbb{W}}_{\psi}^{\epsilon,\kappa} := \mathbb{W}^{\epsilon,\kappa}\left(\hat{\mathbb{P}}_{\psi}^{\mathrm{T}=1}(r), \hat{\mathbb{P}}_{\psi}^{\mathrm{T}=0}(r)\right), \quad (9)$$

where the hard mass-preservation constraint is removed to mitigate the MSE issue. Inspired by Fatras et al. (2021), the robustness of RMPR to sampling effects can be further theoretically investigated in Theorem 3.2, where the effect of mini-batch outliers is upper bounded by a constant.

**Theorem 3.2.** *For empirical distributions $\alpha, \beta$ with n and m units, respectively, adding an outlier $a'$ to $\alpha$ and denoting the disturbed distribution as $\alpha'$, we have*

$$\mathbb{W}^{0,\kappa}(\alpha',\beta) - \mathbb{W}^{0,\kappa}(\alpha,\beta) \le 2\kappa(1 - e^{-\sum_{b\in\beta}(a'-b)^2/2\kappa})/n, \quad (10)$$

*which is upper bounded by $2\kappa/n$. $\mathbb{W}^{0,\kappa}$ is the unbalanced discrepancy as per Definition 3.2.*

In addition, compared with alternatives (Xu et al., 2020; Chapel et al., 2021) to relax the marginal constraint, the approach in Definition 3.2 has better metric properties (Séjourné et al., 2019) and can be accelerated via the generalized Sinkhorn algorithm (Chizat et al., 2018). It is differentiable *w.r.t.* $\psi$ and thus can be minimized via stochastic gradient methods in an end-to-end manner.

### 3.3 PROXIMAL FACTUAL OUTCOME REGULARIZER FOR UNOBSERVED CONFOUNDERS

Existing representation-based methods fail to eliminate the treatment selection bias due to the unobserved confounder effects (UCE). Beginning with CFR (Shalit et al., 2017), the unconfoundedness assumption A.1 (see Appendix A) is often taken to circumvent the UCE issue (Ma et al., 2022). Given two units $r_i \in \mathbb{P}_{\psi}^{T=1}(r)$ and $r_j \in \mathbb{P}_{\psi}^{T=0}(r)$, for instance, optimal transport in Definition 3.2 calculates the unit-wise distance as $\mathbf{D}_{ij} := \|r_i - r_j\|^2$. If Assumption A.1 holds, this approach mitigates the treatment selection bias since it blocks the backdoor

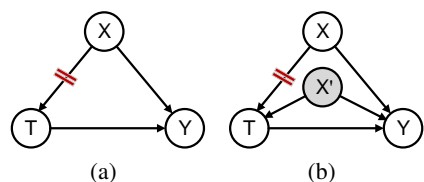

Figure 3: Causal graphs with (a) and w/o (b) the unconfoundedness assumption. The shaded node indicates the hidden confounder $X'$.

path $X \to T$ in Figure 3(a) by balancing the confounders across groups in a latent space. However, Assumption A.1 is usually violated in practice as per Figure 3(b), which hinder existing methods including OT from handling treatment selection bias since the backdoor path $X' \to T$ is not blocked.

According to Figure 3(b), given balanced $X$ and identical $T$, the only variable reflecting the variation of $X'$ is the outcome $Y$. As such, inspired by the joint distribution transport technique (Courty et al., 2017a), we propose to calibrate the unit-wise distance $\mathbf{D}$ with the potential outcomes as follow:

$$\mathbf{D}_{ij}^{\gamma} = \|r_i - r_j\|^2 + \gamma \cdot \left[\|y_i^{T=0} - y_j^{T=0}\|^2 + \|y_j^{T=1} - y_i^{T=1}\|^2\right], \quad (11)$$

where $\gamma$ controls the strength of regularization. The underlying regularization is: units with similar (both observed and unobserved) confounders should have similar potential outcomes. As such, for a pair of units with similar observed covariates, *i.e.*, $\|r_i - r_j\|^2 \approx 0$, if their potential outcomes given the same treatment $t = \{0, 1\}$ differ greatly, *i.e.*, $\|y_i^t - y_j^t\| >> 0$, their unobserved confounders should likewise differ significantly. The vanilla OT technique in (6) with $\mathbf{D}_{ij} = \|r_i - r_j\|^2$ would incorrectly

match this pair because $\|r_i - r_j\|^2 \approx 0$, generate a false transport strategy, and consequently misguide the update of the representation mapping $\psi$. In contrast, OT based on $\mathbf{D}_{ij}^{\gamma}$ would not match this pair as the difference of unobserved confounders is compensated with that of potential outcomes.

Moreover, since $y_i^{T=0}$ and $y_j^{T=1}$ in (11) are unavailable due to the missing counterfactual outcomes, the proposed proximal factual outcome regularizer (PFOR) uses their estimates instead. Specifically, let $\hat{y}_i$ and $\hat{y}_j$ be the estimates of $y_i^{T=0}$ and $y_j^{T=1}$, respectively, PFOR refines (11) as

$$\mathbf{D}_{ij}^{\gamma} = \|r_i - r_j\|^2 + \gamma \cdot \left[ \|\hat{y}_i - y_j\|^2 + \|\hat{y}_j - y_i\|^2 \right], \quad \hat{y}_i = \phi_0(r_i), \quad \hat{y}_j = \phi_1(r_j), \qquad (12)$$

Additional justifications, assumptions and limitations of PFOR are discussed in Appendix D.3.

### 3.4 Architecture of Entire Space Counterfactual Regression

The architecture of ESCFR is presented in Figure 1(b), where the covariate $X$ is first mapped to the representations $R$ with $\psi(\cdot)$, and then to the potential outcomes with $\phi(\cdot)$. The group discrepancy $\mathbb{W}$ is calculated with the optimal transport equipping with the RMPR in (8) and PFOR in (12).

The learning objective is to minimize the risk of factual outcome estimation and the group discrepancy. Given mini-batch distributions $\hat{\mathbb{P}}^{T=1}(x)$ and $\hat{\mathbb{P}}^{T=0}(x)$ in Definition 3.1, the risk of factual outcome estimation following Shi et al. (2019) can be formulated as

$$\mathcal{L}_{\mathrm{F}}(\psi, \phi) := \mathbb{E}_{x_i \in \hat{\mathbb{P}}^{T=1}(x)} \|\phi_1(\psi(x_i)) - y_i\|^2 + \mathbb{E}_{x_j \in \hat{\mathbb{P}}^{T=0}(x)} \|\phi_0(\psi(x_j)) - y_j\|^2, \qquad (13)$$

where $y_i$ and $y_j$ are the factual outcomes for the corresponding treatment groups. The discrepancy is:

$$\mathcal{L}_{\mathrm{D}}^{\epsilon, \kappa, \gamma}(\psi) := \mathbb{W}^{\epsilon, \kappa} \left( \hat{\mathbb{P}}_{\psi}^{\mathrm{T}=1}(r), \hat{\mathbb{P}}_{\psi}^{\mathrm{T}=0}(r) \right), \qquad (14)$$

which is in general the optimal transport with RMPR in Definition 3.2, except for the unit-wise distance $\mathbf{D}^{\gamma}$ calculated with the PFOR in (12). Finally, the overall learning objective of ESCFR is

$$\mathcal{L}_{\mathrm{ESCFR}}^{\epsilon, \kappa, \gamma, \lambda} := \mathcal{L}_{\mathrm{F}}(\psi, \phi) + \lambda \cdot \mathcal{L}_{\mathrm{D}}^{\epsilon, \kappa, \gamma}(\psi), \qquad (15)$$

where $\lambda$ controls the strength of distribution alignment, $\epsilon$ controls the entropic regularization in (5), $\kappa$ controls RMPR in (8), and $\gamma$ controls PFOR in (12). The learning objective (15) mitigates the selection bias following Theorem 3.1 and handles the MSE and UCE issues.

The optimization procedure consists of three steps as summarized in Algorithm 3. First, compute $\boldsymbol{\pi}^{\epsilon, \kappa}$ by solving the linear programming problem in Definition 3.2 with Algorithm 2, where the unit-wise distance is calculated with $\mathbf{D}^{\gamma}$. Second, compute the discrepancy in (14) as $\langle \boldsymbol{\pi}^{\epsilon, \kappa}, \mathbf{D}^{\gamma} \rangle$, where $\mathbf{D}^{\gamma}$ is differentiable to $\psi$, making it feasible to minimize this discrepancy with its gradient w.r.t. $\psi$. Finally, calculate the overall loss in (15) and update $\psi$ and $\phi$ with stochastic gradient methods.

## 4 Experiments

### 4.1 Experimental Setup

**Datasets.** Missing counterfactuals impedes the evaluation of PEHE with observational benchmarks. Following Liuyi et al. (2018); Shalit et al. (2017), experiments are conducted on two semi-synthetic benchmarks. Specifically, the IHDP benchmark aims to estimate the effect of specialist home visits on infants' potential cognitive scores, with 747 observations and 25 covariates; the ACIC dataset comes from the collaborative perinatal project, with 4802 observations and 58 covariates.

**Baselines.** We consider three groups of baselines. (1) Statistical estimators: least square regression with the treatment as covariates (OLS), random forest with the treatment as covariates (R.Forest), a single network with the treatment as covariates (S.learner by Künzel et al. (2019)), separate neural regressors for each treatment group (T.learner by Künzel et al. (2019)), TARNet (Shalit et al., 2017); (2) Matching estimators: propensity score match with logistic regression (PSM by Rosenbaum & Rubin (1983b)), k-nearest neighbor (k-NN by Crump et al. (2008)), causal forest (C.Forest by Wager & Athey (2018)), orthogonal forest (O.Forest by Wager & Athey (2018)); (3) Representation-based methods: balancing neural network (BNN by Johansson et al. (2016)), counterfactual regression with MMD (CFR-MMD) and Wasserstein metric (CFR-WASS) by Shalit et al. (2017).

Table 1: Performance (mean±std) on the PEHE and AUUC metrics. "*" marks the baseline estimators that ESCFR outperforms significantly at p-value < 0.05 over paired samples t-test.

| Dataset | ACIC (PEHE) | | IHDP (PEHE) | | ACIC (AUUC) | | IHDP (AUUC) | |
|---|---|---|---|---|---|---|---|---|
| Model | In-sample | Out-sample | In-sample | Out-sample | In-sample | Out-sample | In-sample | Out-sample |
| OLS | 3.749±0.080* | 4.340±0.117* | 3.856±6.018 | 5.674±9.026 | 0.843±0.007 | 0.496±0.017* | 0.652±0.050 | 0.492±0.032* |
| R.Forest | 3.597±0.064* | 3.399±0.165* | 2.635±3.598 | 4.671±9.291 | 0.902±0.016 | 0.702±0.026* | 0.736±0.142 | 0.661±0.259 |
| S.Learner | 3.572±0.269* | 3.636±0.254* | 1.706±1.600* | 3.038±5.319 | **0.905±0.041** | 0.627±0.014* | 0.633±0.183 | 0.702±0.330 |
| T.Learner | 3.429±0.142* | 3.566±0.248* | 1.567±1.136* | 2.730±3.627 | 0.846±0.019 | 0.632±0.020* | 0.651±0.179 | 0.707±0.333 |
| TARNet | 3.236±0.266* | 3.254±0.150* | 0.749±0.291 | 1.788±2.812 | 0.886±0.046 | 0.662±0.014* | 0.654±0.184 | 0.711±0.329 |
| C.Forest | 3.449±0.101* | 3.196±0.177* | 4.018±5.602* | 4.486±8.677 | 0.717±0.005* | 0.709±0.018* | 0.643±0.141 | 0.695±0.294 |
| k-NN | 5.605±0.168* | 5.892±0.138* | 2.208±2.233* | 4.319±7.336 | 0.892±0.007* | 0.507±0.034* | 0.725±0.142 | 0.668±0.299 |
| O.Forest | 8.094±4.669* | 4.148±2.224* | 2.605±2.418* | 3.136±5.642 | 0.744±0.013 | 0.699±0.022* | 0.664±0.157 | 0.702±0.325 |
| PSM | 5.228±0.154* | 5.094±0.301* | 3.219±4.352* | 4.634±8.574 | 0.884±0.010 | 0.745±0.021 | **0.740±0.149** | 0.681±0.253 |
| BNN | 3.345±0.233* | 3.368±0.176* | 0.709±0.330 | 1.806±2.837 | 0.882±0.033 | 0.645±0.013* | 0.654±0.184 | 0.711±0.329 |
| CFR-MMD | 3.182±0.174* | 3.357±0.321* | 0.777±0.327 | 1.791±2.741 | 0.871±0.032 | 0.659±0.017* | 0.655±0.183 | 0.710±0.329 |
| CFR-WASS | 3.128±0.263* | 3.207±0.169* | 0.657±0.673 | 1.704±3.115 | 0.873±0.029 | 0.669±0.018* | 0.656±0.187 | 0.715±0.329 |
| ESCFR | **2.252±0.297** | **2.316±0.613** | **0.607±0.328** | **1.257±0.677** | 0.796±0.030 | **0.754±0.021** | 0.659±0.187 | **0.734±0.329** |

**Training protocol.**    A fully connected neural network with two 60-dimensional hidden layers is selected to instantiate the representation mapping $\psi$ and the factual outcome mapping $\phi$ for ESCFR and other neural network based baselines. To ensure a fair comparison, all neural models are trained for 400 epochs with Adam optimizer, with the learning rate and weight decay both set to 0.001. Other settings of optimizers follow Kingma & Ba (2015). We fine-tune hyperparameters within the range in Figure 5, validate performance every two epochs, and save the optimal model for test.

**Evaluation protocol.**    Following Liuyi et al. (2018), PEHE in (3) is used as the precision metric for performance evaluation. However, it is unavailable in the model selection phase due to missing counterfactuals. As such, we use the area under the uplift curve (AUUC) (Betlei et al., 2021) to guide model selection, which measures the counterfactual ranking performance of the model and can be computed without counterfactual outcomes. The within-sample and out-of-sample results are reported on the training and test data, respectively, following Shalit et al. (2017).

## 4.2    PERFORMANCE COMPARISON

Table 1 reports the performance of ESCFR and its competitors over ten runs. Statistical estimators exhibit competitive performance on the PEHE metric. Due to the superiority to depict non-linearity, neural estimators outperform linear and random forest methods. In particular, TARNet that absorbs the advantage of T-learner and S-learner achieves the best overall performance in statistic estimators. However, the circumvention to treatment selection bias leads to inferior performance.

Matching methods, *e.g.*, PSM exhibit compelling ranking performance, which explains why they are favored in counterfactual ranking practice (Betlei et al., 2021). However, their poor performance on PEHE hinders their application in counterfactual estimation applications such as advertising systems that place more emphasis on the accuracy of treatment effect estimation.

Representation-based methods mitigate the treatment selection bias and enhance overall performance. In particular, CFR-WASS reaches an out-of-sample PEHE of 3.207 on ACIC, significantly outperforming most statistical methods. However, the MSE and UCE issues impede these methods from solving the treatment selection bias. The proposed ESCFR achieves significant improvement over most metrics compared with various prevalent baselines [4]. Combined with the comparisons above, we attribute its superiority to the proposed RMPR and PFOR regularizers, which makes it robust to MSE and UCE. See Appendix D.5 for additional comparison results.

## 4.3    ABLATION STUDY

To verify the effectiveness of individual components, an ablation study is conducted on the ACIC benchmark in Table 2. Specifically, ESCFR first augments TARNet with stochastic optimal transport in Section 3.1, which effectively reduces the out-of-sample PEHE from 3.254 to 3.207. Then, it

---

[4]An exception would be the within-sample AUUC, which is reported over training data and thus easy to be overfitted. This metric is not critical as the factual outcomes are typically unavailable in the inference phase. We mainly rely on out-of-sample AUUC instead to evaluate the ranking performance and perform model selection.

Table 2: Ablation study (mean±std) on the ACIC benchmark. "*" marks the variants that ESCFR outperforms significantly at p-value < 0.01 over paired samples t-test.

| SOT | RMPR | PFOR | In-sample | | Out-sample | |
| --- | --- | --- | --- | --- | --- | --- |
| | | | PEHE | AUUC | PEHE | AUUC |
| ✗ | ✗ | ✗ | 3.2367±0.2666* | **0.8862+0.0462** | 3.2542+0.1505* | 0.6624+0.0149* |
| ✓ | ✗ | ✗ | 3.1284±0.2638* | 0.8734+0.0291 | 3.2073+0.1699* | 0.6698+0.0187* |
| ✓ | ✓ | ✗ | 2.6459+0.2747* | 0.8356+0.0286 | 2.7688±0.4009 | 0.7099+0.0157* |
| ✓ | ✗ | ✓ | 2.5705±0.3403* | 0.8270±0.0341 | 2.6330±0.4672 | 0.7110±0.0287* |
| ✓ | ✓ | ✓ | **2.2520±0.2975** | 0.7968±0.0307 | **2.3165+0.6136** | **0.7542±0.0202** |

mitigates the MSE issue with RMPR in Section 3.2 and the UCE issue with PFOR in Section 3.3, reducing the out-of-sample PEHE to 2.768 and 2.633, respectively. Finally, ESCFR combines the RMPR and PFOR in a unified framework in Section 3.4, reducing the out-of-sample PEHE to 2.316.

## 4.4 ANALYSIS OF RELAXED MASS-PRESERVING REGULARIZER

Most prevalent methods fail to cope with the label imbalance and mini-batch outliers in Figure 2(b-c). Figure 4 shows the transport plan generated with RMPR in the same situations, where RMPR alleviates the MSE issue in both bad cases. RMPR with $\kappa = 10$, for instance, avoids the incorrect matching of units with different outcomes; RMPR with $\kappa = 2$ gets robust to the outlier's interference and correctly matches the remaining units. We attribute the success to the relaxed mass-preserving constraint in Section 3.2.

Notably, RMPR does not transport all mass of a unit. The closer the unit is to the overlapping zone in a batch, the greater the mass is transferred. That is, RMPR adaptively matches and pulls closer units that are close to the overlapping region, ignoring outliers, which mitigates the bias of causal inference methods in cases where the positivity assumption does not strictly hold. Current approaches mainly achieve it by manually cleaning the data or dynamically weighting the units (Johansson et al., 2020), while RMPR naturally implements it via the soft penalty in (8).

We further investigate the performance of RMPR under different batch sizes and $\kappa$ in Appendix D.2 to verify the effectiveness of RMPR more thoroughly.

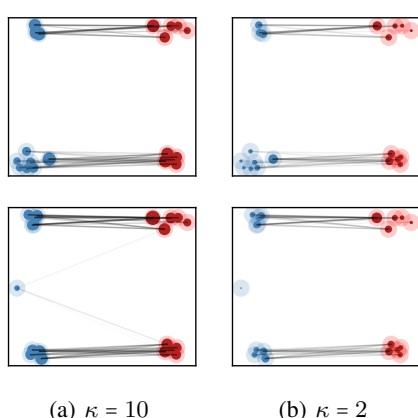

(a) $\kappa = 10$      (b) $\kappa = 2$

Figure 4: Geometric interpretation of optimal transport plan with RMPR under the outcome imbalance (upper) and outlier (down) settings. The dark area indicates the transported mass of a unit, i.e., marginal of the transport matrix $\pi$. The light area indicates the total mass.

## 4.5 PARAMETER SENSITIVITY STUDY

We discuss four critical hyperparameters in ESCFR, i.e., $\lambda$, $\epsilon$, $\kappa$, $\gamma$, which are the weights in the learning objective and influence the final performance significantly.

We firstly vary $\lambda$ to investigate the influence of stochastic optimal transport. Specifically, increasing $\lambda$ consistently improves the precision of ITE estimates. For example, the out-of-sample PEHE reduces from 3.22 at $\lambda = 0$ to approximately 2.85 at $\lambda = 1.5$. However, assigning a higher focus on distribution balancing in a multi-task learning framework leads to difficulties in factual outcome estimation, leading to sub-optimal ITE estimates. Then, we vary $\epsilon$ to investigate the entropic regularizer. It does accelerate the computation of optimal transport discrepancy with a large $\epsilon$ corresponding to a faster computation speed (Flamary et al., 2021). However, it comes with a biased transport plan, evidenced by the out-of-sample PEHE increasing to 2.95 with fluctuations.

We further vary $\gamma$ and $\kappa$ to showcase the influence of PFOR and RMPR, respectively. PFOR benefits ITE estimation, while assigning considerable weight to the proximal outcome distance is detrimental to ITE estimates, because the unit-wise distance calculated with representations (covariates) would be neglected. Relaxing the mass-preserving constraint via RMPR could significantly improve the ITE estimates; however, too-small $\kappa$ is always detrimental as we cannot guarantee that the representations across treatment groups are drawn closer together in the optimal transport approach.

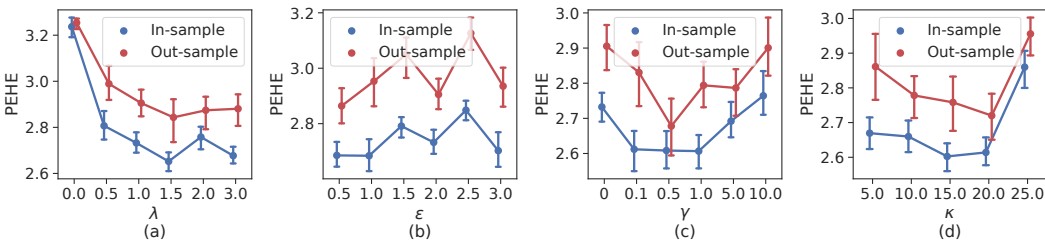

Figure 5: Parameter sensitivity study for critical hyper-parameters of ESCFR

## 5 RELATED WORKS

Current works mitigate the treatment selection bias by balancing the distributions of treated and untreated groups, which can be divided into three groups according to the methods used for balancing.

Reweighting-based methods weight individuals with *balanced scores* to achieve globally balanced distributions, represented by the inverse propensity score (IPS) approach (Rosenbaum & Rubin, 1983a) and its doubly robust variant (Robins et al., 1994). Imai & Ratkovic (2014) and Fong et al. (2018) propose calculating the balancing score by solving an optimization problem. Kuang et al. (2017b) and Kuang et al. (2017a) consider additional non-confounding factors in covariates. However, these methods are susceptible to non-overlapping units and suffer from a high variance issue.

Matching-based methods match similar units from different groups to construct locally balanced distributions. Representative methods (Rosenbaum & Rubin, 1983b; Chang & Dy, 2017; Li et al., 2016) mainly differ in terms of similarity measures. Notably, Tree-based methods (Wager & Athey, 2018) can also be considered as matching methods with adaptive neighborhood metrics. However, the computational cost hinders the application of these methods in large-scale scenarios.

Begining with BNN (Johansson et al., 2016) and CFR (Shalit et al., 2017), representation-based methods minimize the group discrepancy in the latent space. Liuyi et al. (2018) and Hassanpour & Greiner (2020) further augment CFR with local similarity and non-confounding factors, respectively. Kallus (2020) and Yoon et al. (2018) propose to balance the distributions of representations with adversarial training. Due to its scalability and avoidance of the high variance issue, representation-based methods have been predominant for handling the treatment selection bias.

It is necessary to distinguish ourselves from emerging OT-based causal inference approaches. Dunipace (2021) augments the IPS method with a propensity score estimator based on OT; however, it is limited by the aforementioned high variance issue. Torous et al. (2021) uses the push-forward operator to improve change-in-change models; however, they are designed for multi-phase data which is not available in our case. Li et al. (2022) has a similar setup to to us, while it focuses on the decomposition of latent variables and is identical to Shalit et al. (2017) in terms of alignment technology. Our work is a new take on OT under the CFR framework, alleviating the MSE and UCE issues that have been long neglected by the causal inference community until this year.

## 6 CONCLUSION

Due to the effectiveness of mitigating treatment selection bias, representation learning has been the primary approach to estimating individual treatment effect. However, existing methods neglect the mini-match sampling effects and unobserved confounders, which hinders them from handling the treatment selection bias. A principled approach named ESCFR is devised based on a generalized Sinkhorn discrepancy. Extensive experiments demonstrate that ESCFR can largely mitigate MSE and UCE issues, and achieve better performance compared with other baseline models.

There are two directions of future work that we intend to pursue. The first direction attempts to construct the representation mapping with normalizing flows (Chen et al., 2018), which is invertible and thus satisfies the assumption to the representation mapping by Shalit et al. (2017). Another direction seeks to extend our methodology to industrial applications, *e.g.*, debias in recommenders (Wang et al., 2022), which has long been dominated by reweighting methods with the high variance issue.

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

## A  CAUSAL INFERENCE WITH OBSERVATIONAL STUDIES

We introduce all the necessary preliminaries about causal inference for readers and reviewers that are unfamiliar with this area.

### A.1  PROBLEM FORMULATION

This section formalizes the definitions, assumptions, and useful lemmas in causal inference from observational data. Following the notations in Section 2.1, an individual with covariates $x$ has two potential outcomes, namely $Y_1(x)$ given it is treated and $Y_0(x)$ otherwise. The ground-truth individual treatment effect (ITE) is the difference in its potential outcomes.

**Definition A.1.** *The individual treatment effect (ITE) for a unit with covariates $x$ is*

$$\tau(x) := \mathbb{E}\left[Y_1 - Y_0 \mid x\right], \tag{16}$$

*where we abbreviate $Y_1(x)$ to $Y_1$ for brevity. The expectation is over the potential outcome space $\mathcal{Y}$.*

Estimating ITE with observational data is a common practice in causal inference, which has long been confronted with two primary challenges:

- Missing counterfactuals: where only the factual outcome is observable. If a patient is treated, for instance, we can never observe what would have happened if the patient was untreated in the same situation.
- Treatment selection bias, where individuals have preferences for treatment selection. For example, doctors would adapt different treatment plans for patients with different health conditions. It would make the treated and untreated populations heterogeneous. ITE estimators naïvely trained to minimize the factual outcome error would overfit the respective group's properties and thus cannot generalize well to the entire population.

Pearl & Mackenzie (2018) suggested a two-step methodology to overcome these two challenges. The first step is identification, which aims to construct an unbiased statistical estimand to identify the causal estimand (*e.g.*, $\tau(x)$) based on the adjustment formula. Note that not all causal estimands are identifiable, *e.g.*, ITE is identifiable only if Assumption A.1-A.3 hold.

**Assumption A.1.** *(Unconfoundedness). For all covariates $x$ in the population of interest (i.e., $x$ with $\mathbb{P}(X = x) > 0$), we have conditional independence $(Y_0, Y_1) \perp\!\!\!\perp T \mid X = x$. That is, potential outcomes are conditionally independent of treatment assignment.*

**Assumption A.2.** *(Consistency). For all covariates $x$ in the population of interest, we have $Y = Y_t$. That is, the observed outcome is consistent with the potential outcome w.r.t. the assigned treatment.*

**Assumption A.3.** *(Positivity). For all covariates $x$ in the population of interest, we have $0 < \mathbb{P}(T = 1 \mid X = x) < 1$. That is, all individuals have a chance to be assigned both treatments.*

The second step is estimation, which aims to estimate the derived statistical estimand with observational data. Lemma A.1 illustrates how this two-step approach can be used for ITE estimation.

**Lemma A.1.** *The ITE estimand $\tau(x)$ can be identified as:*

$$\mathbb{E}\left[Y_1 - Y_0 \mid X = x\right] = \mathbb{E}\left[Y_1 \mid X = x\right] - \mathbb{E}\left[Y_0 \mid X = x\right]$$
$$\overset{(1)}{=} \mathbb{E}\left[Y_1 \mid X = x, T = 1\right] - \mathbb{E}\left[Y_0 \mid X = x, T = 0\right] \tag{17}$$
$$\overset{(2)}{=} \mathbb{E}\left[Y \mid X = x, T = 1\right] - \mathbb{E}\left[Y \mid X = x, T = 0\right],$$

*where (1) stems from the unconfoundedness assumption A.1; (2) stems from the consistency assumption A.2. The derived estimand is fully composed of statistical estimands, which can only be estimated under the positivity assumption A.3. Otherwise, if the positivity assumption is violated, we have:*

$$\mathbb{E}\left[Y \mid X = x, T = 1\right] = \int y \cdot \mathbb{P}(Y = y \mid X = x, T = 1)\, dy$$
$$= \int y \cdot \frac{\mathbb{P}(Y = y, X = x, T = 1)}{\mathbb{P}(T = 1 \mid X = x)\mathbb{P}(X = x)}\, dy, \tag{18}$$

*which is not computable as there exists $x \in \mathcal{X}$ which makes $\mathbb{P}(T = 1 \mid X = x) = 0$.*

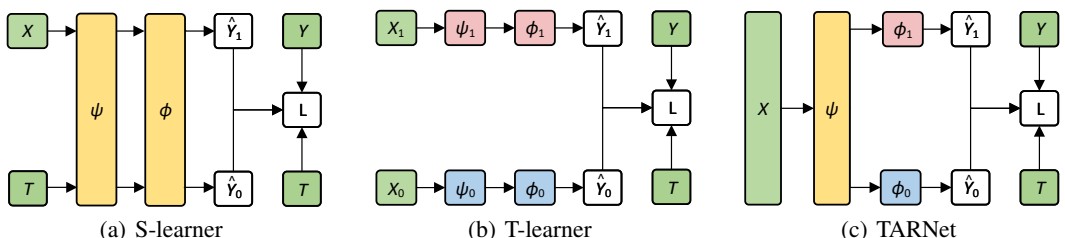

|(a) S-learner|(b) T-learner|(c) TARNet|

Figure 6: Architecture of Meta-learner based ITE estimators, consisting of inputs (green), outputs (white), shared mappings (yellow), and mappings for treated and untreated units (red and blue, respectively).

## A.2 META-LEARNERS FOR ITE ESTIMATION WITH OBSERVATIONAL DATA

In an effort to solve missing counterfactuals, existing meta-learner based methods (Künzel et al., 2019; Nie & Wager, 2020) decompose the ITE estimation problem into several subproblems that can be solved with any supervised learning method. As depicted in Figure 6, S-learner regards the treatment indicator $T$ as one of the covariates $X$, and utilizes the shared representation mapping $\psi$ and outcome mapping $\phi$ to estimate the factual outcomes. However, because the network structure does not highlight the role of treatment indicator, it may be overlooked when treatment effects are minimal. T-learner models the factual outcomes for treated units $X_1$ and untreated units $X_0$ separately, which highlights the treatment indicator's effect; however, it reduces the data efficiency and is therefore inapplicable when the dataset is small. Künzel et al. (2019) discuss the advantages and limitations of these two approaches in more detail.

**Definition A.2.** *Let $\psi : \mathcal{X} \to \mathcal{R}$ be a mapping from support $\mathcal{X}$ to $\mathcal{R}$. That is, $\forall x \in \mathcal{X}$, $\exists r = \psi(x) \in \mathcal{R}$. Let $\phi : \mathcal{R} \times \mathcal{T} \to \mathcal{Y}$ be a mapping from support $\mathcal{R} \times \mathcal{T}$ to $\mathcal{Y}$. That is, it maps the representations and treatment indicator to the corresponding factual outcome. For example, $Y_1 = \phi_1(R)$, $Y_0 = \phi_0(R)$, where we will always abbreviate $\phi(R, T = 1)$ and $\phi(R, T = 0)$ to $\phi_1(R)$ and $\phi_0(R)$, respectively, for brevity.*

**Assumption A.4.** *$\phi : \mathcal{X} \to \mathcal{R}$ is differentiable and invertible, with its inverse $\phi^{-1}$ defined over $\mathcal{R}$.*

TARNet (Shalit et al., 2017) in Figure 6 (c) obtains better results by absorbing the advantages of both T-learner and S-learner, which consists of a representation mapping $\psi$ and an outcome mapping $\phi$ as defined in Definition A.2. For a unit with covariates $X$, TARNet estimates ITE as the difference in predicted outcomes when $T$ is set to treated and untreated:

$$\hat{\tau}_{\psi,\phi}(X) \coloneqq \hat{Y}_1 - \hat{Y}_0, \quad \text{where} \quad \hat{Y}_1 = \phi_1(\psi(X)), \quad \hat{Y}_0 = \phi_0(\psi(X)), \tag{19}$$

where $\psi$ is trained over all units, $\phi_1$ and $\phi_0$ are trained over the treated and untreated units, respectively, to minimize the factual error $\epsilon_F(\phi, \psi)$ in Definition A.3. Finally, the performance of the ITE estimator is mainly evaluated with PEHE:

$$\epsilon_{\text{PEHE}}(\psi, \phi) = \int_{\mathcal{X}} \left( \hat{\tau}_{\psi,\phi}(x) - \tau(x) \right)^2 \mathbb{P}(x)\, dx. \tag{20}$$

**Definition A.3.** *Let $\mathbb{L}$ be the loss function that measures the quality of outcome estimation, e.g., the squared loss. The expected loss for the units with covariates $x$ and treatment indicator $t$ is:*

$$l_{\psi,\phi}(x, t) \coloneqq \int_{\mathcal{Y}} \mathbb{L}(Y_t, \phi(\psi(x), t)) \cdot \mathbb{P}(Y_t \mid x)\, dY_t. \tag{21}$$

*where $\mathbb{L}$ is realized with the squared loss: $\mathbb{L}(Y_t, \psi(\phi(x), t)) = (Y_t - \psi(\phi(x), t))^2$ in our scenario. The expected factual outcome estimation error for treated, untreated and all units are:*

$$\epsilon_F^{T=1}(\psi, \phi) \coloneqq \int_{\mathcal{X}} l_{\psi,\phi}(x, 1) \cdot \mathbb{P}^{T=1}(x)\, dx,$$

$$\epsilon_F^{T=0}(\psi, \phi) \coloneqq \int_{\mathcal{X}} l_{\psi,\phi}(x, 0) \cdot \mathbb{P}^{T=0}(x)\, dx, \tag{22}$$

$$\epsilon_F(\psi, \phi) \coloneqq \int_{\mathcal{X} \times \mathcal{T}} l_{\psi,\phi}(x, t) \cdot \mathbb{P}(x, t)\, dx dt.$$

However, the treatment selection bias makes covariate distributions across groups shift. As such, $\phi_1$ and $\phi_0$ would overfit the respective group's properties and thus cannot generalize well to the entire population. For example, as shown in Figure 1(a) (a), the potential outcome estimator $\phi_1$ trained with treated units cannot generalize to the untreated units. Therefore, the resulting $\hat{\tau}$ would be biased.

**Definition A.4.** *Let* $\mathbb{P}^{T=1}(x) := \mathbb{P}(x \mid T = 1)$ *and* $\mathbb{P}^{T=0}(x) := \mathbb{P}(x \mid T = 0)$ *be the covariate distribution for treated and untreated groups, respectively. Let* $\mathbb{P}_{\psi}^{T=1}(r)$ *and* $\mathbb{P}_{\psi}^{T=0}(x)$ *be that of representations induced by the representation mapping* $r = \psi(x)$ *defined in Definition 2.2.*

To mitigate the effect of treatment selection bias, representation-based approaches (Johansson et al., 2016; Shalit et al., 2017) minimize the distribution discrepancy of different groups in the representation space. In particular, the integral probability metric (IPM) in Definition A.4 is a widely used metric that measures the discrepancy of two distributions. Shalit et al. (2017) propose to optimize the PEHE by minimizing the estimation error of factual outcomes $\epsilon_F$ and the IPM of learned representations between treated and untreated groups. They further provide theoretical results to back up their claim as per Theorem A.1.

**Definition A.5.** *Consider two distribution functions* $\mathbb{P}^{T=1}(x)$ *and* $\mathbb{P}^{T=0}(x)$ *supported over* $\mathcal{X}$, *let* $\mathcal{F}$ *be a sufficiently large function family, the integral probability metric induced by* $\mathcal{F}$ *is*

$$\text{IPM}_{\mathcal{F}}\left(\mathbb{P}^{T=1}, \mathbb{P}^{T=0}\right) = \sup_{f \in \mathcal{F}} \left| \int_{\mathcal{X}} f(x) \left(\mathbb{P}^{T=1}(x) - \mathbb{P}^{T=0}(x)\right) dx \right|, \tag{23}$$

**Theorem A.1.** *Let* $\psi$ *and* $\phi$ *be the mappings in Definition 2.2,* $\mathcal{F}$ *be a predefined sufficiently large function family of* $\phi$, $\text{IPM}_{\mathcal{F}}$ *be the integral probability metric induced by* $\mathcal{F}$. *Assume there exists a constant* $B_{\psi} > 0$, *such that for* $t \in \{0, 1\}$, $\frac{1}{B_{\psi}} \cdot l_{\psi,\phi}(x, t) \in \mathcal{F}$ *holds. Shalit et al. (2017) demonstrate:*

$$\epsilon_{\text{PEHE}}(\psi, \phi) \leq 2\left(\epsilon_F^{T=0}(\psi, \phi) + \epsilon_F^{T=1}(\psi, \phi) + B_{\psi}\text{IPM}_{\mathcal{F}}\left(\mathbb{P}_{\psi}^{T=1}, \mathbb{P}_{\psi}^{T=0}\right) - 2\sigma_Y^2\right), \tag{24}$$

*where* $\epsilon_F^{T=0}$ *and* $\epsilon_F^{T=1}$ *follow Definition A.3,* $\mathbb{P}_{\psi}^{T=1}(r)$ *and* $\mathbb{P}_{\psi}^{T=0}(x)$ *follow Definition A.4.*

## A.4   THEORETICAL RESULTS AND EXTENSIONS

In this section, we describe the intuition of Theorem 3.1 and Theorem 3.2, and provide rigorous proofs to support our claims. Two weaknesses hinder the existing Theorem A.1 by Shalit et al. (2017) from supporting representation-based causal inference approaches. Firstly, the IPM is a discrepancy metric with profound theoretical properties while being difficult to compute numerically. To counter this, note that the IPM holds for any sufficiently large function families, *e.g.*, the 1-Lipschitz function family where the IPM is equivalent to the Wasserstein divergence as per Kantorovich-Rubinstein duality (Villani, 2009). As such, a shortcut to IPM is the Wasserstein distance as per Lemma A.2.

**Lemma A.2.** *Consider two distribution functions* $\mathbb{P}_1(x)$ *and* $\mathbb{P}_2(x)$ *supported over* $\mathcal{X}$; *let* $\mathcal{F}$ *be the family of* 1-*Lipschitz functions,* $\mathbb{W}$ *be the Wasserstein distance, Villani (2009) demonstrates*

$$\text{IPM}_{\mathcal{F}}\left(\mathbb{P}_1, \mathbb{P}_2\right) = \mathbb{W}\left(\mathbb{P}_1, \mathbb{P}_2\right) \tag{25}$$

Another weakness is that it neglects the Mini-batch Sampling Effects (MSE). Specifically, Theorem A.1 holds only if the entire populations of treated and untreated groups are available. However, since the representation-based approaches update parameters with stochastic gradient methods, only a subset of the population is accessible within each iteration. As such, it remains questionable whether Theorem A.1 holds at a mini-batch level in practice.

**Lemma A.3.** *Let* $\mathbb{P}(x)$ *be a probability measure supported over* $\mathcal{X} \in \mathbb{R}^d$ *satisfying* $T_1(\lambda)$ *inequality. Let* $\hat{\mathbb{P}}(x) = \frac{1}{N} \sum_{i=1}^{N} \delta_{x_i}$ *be the corresponding empirical measure with* $N$ *units. Bolley et al. (2007) and Redko et al. (2017) demonstrate that for any* $d' > d$ *and* $\lambda' < \lambda$, *there exists some constant* $N_0$, *such that for any* $\varepsilon > 0$ [5] *and* $N \geq N_0 \max(\varepsilon^{-(d+2)}, 1)$, *we have*

$$\mathbb{P}\left(\mathbb{W}\left(\mathbb{P}(x), \hat{\mathbb{P}}(x)\right) > \varepsilon\right) \leq \exp\left(-\frac{\lambda'}{2}N\varepsilon^2\right) \tag{26}$$

*where* $d', \lambda'$ *can be calculated explicitly.*

---

[5] While there is a risk of symbol reuse, we use $\varepsilon$ here to denote sampling error, and $\epsilon$ to control the strength of entropic regularization in optimal transport.

Hoeffding's inequality is a powerful statistical tool to quantify such sampling effects, which is proved to be applicable for $\mathbb{W}$ by Bolley et al. (2007). Therefore, it is natural to expand $\mathbb{W}$ according to Lemma A.3 to extend Theorem A.1 to mini-batch situations, in order to quantify the sampling effects.

**Theorem A.2.** *Let $\psi$ and $\phi$ be the representation mapping and factual outcome mapping, respectively; $\hat{\mathbb{W}}_\psi$ be the discrepancy across groups at a mini-batch level. With the probability of at least $1 - \delta$, we have:*

$$\epsilon_{\mathrm{PEHE}}(\psi, \phi) \leq 2 \left[ \epsilon_{\mathrm{F}}^{\mathrm{T}=1}(\psi, \phi) + \epsilon_{\mathrm{F}}^{\mathrm{T}=0}(\psi, \phi) + B_\psi \hat{\mathbb{W}}_\psi - 2\sigma_Y^2 + \mathcal{O}(\frac{1}{\delta N}) \right], \qquad (27)$$

*where $\epsilon_{\mathrm{F}}^{\mathrm{T}=1}$ and $\epsilon_{\mathrm{F}}^{\mathrm{T}=0}$ are the expected losses of factual outcome estimation over treated and untreated units, respectively. $N$ is the batch size, $\sigma_Y^2$ is the variance of outcomes, $B_\psi$ is some constant such that $\frac{1}{B_\psi} \cdot l_{\psi,\phi}(x, t)$ belongs to the family of 1-Lipschitz functions, $\mathcal{O}(\cdot)$ is the sampling complexity term.*

*Proof.* According to Theorem A.1 we have:

$$\epsilon_{\mathrm{PEHE}}(\psi, \phi) \leq 2 \left( \epsilon_{\mathrm{F}}^{\mathrm{T}=0}(\psi, \phi) + \epsilon_{\mathrm{F}}^{\mathrm{T}=1}(\psi, \phi) + B_\psi \mathrm{IPM}_{\mathcal{F}} \left( \mathbb{P}_\psi^{\mathrm{T}=1}, \mathbb{P}_\psi^{\mathrm{T}=0} \right) - 2\sigma_Y^2 \right). \qquad (28)$$

Assuming that there exists a constant $B_\psi > 0$, such that for $t \in \{0, 1\}$, $\frac{1}{B_\psi} \cdot l_{\psi,\phi}(x, t)$ belongs to the family of 1-Lipschitz functions. According to Lemma A.2, we have

$$\epsilon_{\mathrm{PEHE}}(\psi, \phi) \leq 2 \left( \epsilon_{\mathrm{F}}^{\mathrm{T}=0}(\psi, \phi) + \epsilon_{\mathrm{F}}^{\mathrm{T}=1}(\psi, \phi) + B_\psi \mathbb{W} \left( \mathbb{P}_\psi^{\mathrm{T}=1}, \mathbb{P}_\psi^{\mathrm{T}=0} \right) - 2\sigma_Y^2 \right). \qquad (29)$$

Following Definition 3.1, let $\hat{\mathbb{P}}_\psi^{\mathrm{T}=1}(r)$ and $\hat{\mathbb{P}}_\psi^{\mathrm{T}=0}(r)$ be the empirical distributions of representations at a mini-batch level, containing $N_1$ treated units and $N_0$ untreated units, respectively. Then we have:

$$\begin{aligned}
\mathbb{W} \left( \mathbb{P}_\psi^{\mathrm{T}=1}, \mathbb{P}_\psi^{\mathrm{T}=0} \right) &\leq \mathbb{W} \left( \mathbb{P}_\psi^{\mathrm{T}=1}, \hat{\mathbb{P}}_\psi^{\mathrm{T}=1} \right) + \mathbb{W} \left( \mathbb{P}_\psi^{\mathrm{T}=0}, \hat{\mathbb{P}}_\psi^{\mathrm{T}=1} \right) \\
&\leq \mathbb{W} \left( \mathbb{P}_\psi^{\mathrm{T}=1}, \hat{\mathbb{P}}_\psi^{\mathrm{T}=1} \right) + \mathbb{W} \left( \mathbb{P}_\psi^{\mathrm{T}=0}, \hat{\mathbb{P}}_\psi^{\mathrm{T}=0} \right) + \mathbb{W} \left( \hat{\mathbb{P}}_\psi^{\mathrm{T}=0}, \hat{\mathbb{P}}_\psi^{\mathrm{T}=1} \right) \\
&:= \mathbb{W} \left( \mathbb{P}_\psi^{\mathrm{T}=1}, \hat{\mathbb{P}}_\psi^{\mathrm{T}=1} \right) + \mathbb{W} \left( \mathbb{P}_\psi^{\mathrm{T}=0}, \hat{\mathbb{P}}_\psi^{\mathrm{T}=0} \right) + \hat{\mathbb{W}}_\psi,
\end{aligned} \qquad (30)$$

because we have the triangular inequality for $\mathbb{W}$. The Hoeffding inequality in Lemma A.3 further gives the following inequality which holds with the probability at least $1 - \delta$:

$$\begin{aligned}
\mathbb{W} \left( \mathbb{P}_\psi^{\mathrm{T}=1}, \hat{\mathbb{P}}_\psi^{\mathrm{T}=1} \right) &\leq \sqrt{2 \log \left( \frac{1}{\delta} \right) / \lambda' N_1} \\
\mathbb{W} \left( \mathbb{P}_\psi^{\mathrm{T}=0}, \hat{\mathbb{P}}_\psi^{\mathrm{T}=0} \right) &\leq \sqrt{2 \log \left( \frac{1}{\delta} \right) / \lambda' N_0}.
\end{aligned} \qquad (31)$$

Denote $N := N_0 + N_1$ as the batch size, $\theta := N_1/N$ as the ratio of treated units in the current batch. Combining (30) and 31 we have

$$\begin{aligned}
\mathbb{W} \left( \mathbb{P}_\psi^{\mathrm{T}=1}, \mathbb{P}_\psi^{\mathrm{T}=0} \right) &\leq \hat{\mathbb{W}}_\psi + \sqrt{2 \log \left( \frac{1}{\delta} \right) / \lambda' N_1} + \sqrt{2 \log \left( \frac{1}{\delta} \right) / \lambda' N_0} \\
&= \hat{\mathbb{W}}_\psi + \sqrt{2 \log \left( \frac{1}{\delta} \right) / \lambda' N} \left( \sqrt{\frac{1}{\theta}} + \sqrt{\frac{1}{1-\theta}} \right) \\
&:= \hat{\mathbb{W}}_\psi + \mathcal{O}(\frac{1}{\delta N}),
\end{aligned} \qquad (32)$$

where $\mathcal{O}(\cdot)$ satisfies

$$\sqrt{\log \left( \frac{1}{\delta} \right) / \lambda'} \left( 1 + \sqrt{1/(N-1)} \right) \geq \mathcal{O}(\frac{1}{\delta N}) \geq 4 \sqrt{\log \left( \frac{1}{\delta} \right) / \lambda' N}, \qquad (33)$$

where $\mathcal{O}(\frac{1}{\delta N})$ reaches its maximum when $\theta = 1/N$ or $\theta = 1 - 1/N$, reaches its minimum when $\theta = 0.5$. This corollary can be derived by differentiating the function $f(x) = 1/\sqrt{x} + 1/\sqrt{1-x}$.

Combining (29) and 32 we have

$$\epsilon_{\mathrm{PEHE}}(\psi, \phi) \leq 2 \left[ \epsilon_{\mathrm{F}}^{\mathrm{T}=1}(\psi, \phi) + \epsilon_{\mathrm{F}}^{\mathrm{T}=0}(\psi, \phi) + B_\psi \hat{\mathbb{W}}_\psi - 2\sigma_Y^2 + \mathcal{O}(\frac{1}{\delta N}) \right], \qquad (34)$$

where we denote $B_\psi \mathcal{O}(\frac{1}{\delta N})$ as $\mathcal{O}(\frac{1}{\delta N})$ and the proof is completed. $\qquad\qquad \square$

Theorem A.2 extends Theorem A.1 and derives the upper bound of PEHE in the stochastic batch form, which demonstrates that the PEHE can be optimized by iteratively minimizing the factual outcome estimation error and the optimal transport discrepancy *at a mini-batch level*.

**Corollary A.1.** *The empirical variance of the PEHE estimates in* (27) *largely depends on the batch size and the proportion of treated and untreated units. Large batch size and balanced proportion produce low empirical variance, and vice versa.*

*Proof.* It can be drawn directly from (27) (batch size) and (33) (treatment proportion). □

**Theorem A.3.** *For discrete measures $\alpha = \sum_{i=1}^{n} \mathbf{a}_i \delta_{\mathbf{x}_i}$ and $\beta = \sum_{j=1}^{m} \mathbf{b}_j \delta_{\mathbf{x}_j}$, adding an outlier $\delta_{\mathbf{x}'}$ to $\alpha$ and denote the disturbed distribution as $\alpha'$, we have*

$$\mathbb{W}^{0,\kappa}\left(\alpha', \beta\right) - \mathbb{W}^{0,\kappa}\left(\alpha, \beta\right) \le 2\kappa\big(1 - e^{-\sum_{j=1}^{m}(\mathbf{x}'-\mathbf{x}_j)^2/2\kappa}\big)/n, \tag{35}$$

*which is upper bounded by $2\kappa/(n+1)$. $\mathbb{W}^{0,\kappa}$ is the unbalanced discrepancy as per Definition 3.2.*

*Proof.* This is a direct corollary of the Lemma 1 by Fatras et al. (2021), under the assumption that all the units including the outlier $\delta_{\mathbf{x}'}$ share the same mass. □

## B DISCRETE OPTIMAL TRANSPORT

This section proposes the definitions and algorithms to calculate optimal transport between discrete measures. We have omitted the case of general measures, as it is beyond the scope of this work. Readers interested in this topic should refer to Peyré & Cuturi (2019); Cuturi (2013) for details.

### B.1 PROBLEM FORMULATION

Optimal transport derives from the formulation of Monge (1781). We provide an equivalent interpretation under discrete measures. Consider $n$ warehouses and $m$ factories, where the $i$-th warehouse contains $\mathbf{a}_i$ units of materials; the $j$-th factory needs $\mathbf{b}_j$ units of materials. Now construct a *mapping* from warehouses to factories, satisfying: (1) all materials of warehouses are transported; (2) all requirements of factories are satisfied; (3) materials from one warehouse are transported to *no more than one* factory (mapping constraint). Every feasible mapping is associated with a *global* cost, calculated by aggregating the *local* cost of moving a unit of material from the $i$-th warehouse to the $j$-th factory. Our objective, to find a feasible mapping that minimizes the transport cost, is formulated in Definition B.1.

**Definition B.1.** *For discrete measures $\alpha = \sum_{i=1}^{n} \mathbf{a}_i \delta_{\mathbf{x}_i}$ and $\beta = \sum_{j=1}^{m} \mathbf{b}_j \delta_{\mathbf{x}_j}$, the Monge problem seeks for a mapping $\mathbb{T} : \{\mathbf{x}_i\}_{i=1}^{n} \to \{\mathbf{x}_j\}_{j=1}^{m}$ that associates to each point $\mathbf{x}_i$ a single point $\mathbf{x}_j$ and pushes the mass of $\alpha$ to $\beta$. That is, $\forall j \in \{1, \ldots, m\}$ we have $\mathbf{b}_j = \sum_{i:\mathbb{T}(\mathbf{x}_i)=\mathbf{x}_j} \mathbf{a}_i$. This mass-preserving constraint is abbreviated as $\mathbb{T}_\sharp \alpha = \beta$. The mapping should also minimize the transportation cost denoted as $c(x, y)$. To this end, Monge problem for discrete measures is formulated as:*

$$\min_{\mathbb{T}:\mathbb{T}_\sharp \alpha=\beta} \left\{ \sum_i c(\mathbf{x}_i, \mathbb{T}(\mathbf{x}_i)) \right\}. \tag{36}$$

This problem was further utilized to compare two probability measures where $\sum_i \mathbf{a}_i = \sum_j \mathbf{b}_j = 1$. However, Monge's formulation cannot guarantee the existence and uniqueness of solutions (Peyré & Cuturi, 2019). Kantorovich (2006) relaxed the mapping constraint by allowing the transport from one warehouse to many factories and reformulated the Monge problem as a linear programming problem in Definition B.2.

**Definition B.2.** *For discrete measures $\alpha = \sum_{i=1}^{n} \mathbf{a}_i \delta_{\mathbf{x}_i}$ and $\beta = \sum_{j=1}^{m} \mathbf{b}_j \delta_{\mathbf{x}_j}$, the Kantorovich problem aims to find a feasible plan $\pi \in \mathbb{R}_+^{n \times m}$ which transports $\alpha$ to $\beta$ at minimum cost:*

$$\mathbb{W}(\alpha, \beta) := \min_{\boldsymbol{\pi} \in \Pi(\alpha, \beta)} \langle \mathbf{D}, \boldsymbol{\pi} \rangle, \quad \Pi(\alpha, \beta) := \left\{ \boldsymbol{\pi} \in \mathbb{R}_+^{n \times m} : \boldsymbol{\pi} \mathbf{1}_m = \mathbf{a}, \boldsymbol{\pi}^\mathsf{T} \mathbf{1}_n = \mathbf{b} \right\}, \tag{37}$$

---

**Algorithm 1** Sinkhorn Algorithm

---

**Input**: discrete measures $\alpha = \sum_{i=1}^{n} \mathbf{a}_i \delta_{\mathbf{x}_i}$ and $\beta = \sum_{j=1}^{m} \mathbf{b}_j \delta_{\mathbf{x}_j}$, distance matrix $\mathbf{D}_{ij} = \|\mathbf{x}_i - \mathbf{x}_j\|_2^2$.
**Parameter**: $\epsilon$: strength of entropic regularization; $\ell_{\max}$: maximum iterations.
**Output**: $\boldsymbol{\pi}^\epsilon$: the entropic regularized optimal transport matrix.

1: $\mathbf{K} \leftarrow \exp(-\mathbf{D}/\epsilon)$.
2: $\mathbf{u} \leftarrow \mathbf{1}_n. \; \mathbf{v} \leftarrow \mathbf{1}_m, \ell \leftarrow 1$.
3: **while** $\ell < \ell_{\max}$ **do**
4:      $\mathbf{u} \leftarrow \mathbf{a}/(\mathbf{Kv})$.
5:      $\mathbf{v} \leftarrow \mathbf{b}/(\mathbf{K}^T\mathbf{u})$.
6:      $\ell \leftarrow \ell + 1$.
7: $\boldsymbol{\pi}^\epsilon \leftarrow \operatorname{diag}(\mathbf{u})\mathbf{K}\operatorname{diag}(\mathbf{v})$.

---

*where $\mathbb{W}(\alpha, \beta) \in \mathbb{R}$ is the Wasserstein discrepancy between $\alpha$ and $\beta$; $\mathbf{D} \in \mathbb{R}_+^{n \times m}$ is the unit-wise distance[6] between $\alpha$ and $\beta$; $\mathbf{a}$ and $\mathbf{b}$ indicate the mass of units in $\alpha$ and $\beta$, and $\Pi$ is the feasible transportation plan set which ensures the mass-preserving constraint holds.*

## B.2 SINKHORN DISCREPANCY AND ALGORITHM

Exact solutions to the Kantorovich problem suffer from great computational costs. The interior-point and network-simplex methods, for example, have a complexity of $\mathcal{O}(n^3 \log n)$ (Pele & Werman, 2009). A shortcut is to add an entropic regularizer as

$$\mathbb{W}^\epsilon(\alpha, \beta) := \langle \mathbf{D}, \boldsymbol{\pi}^\epsilon \rangle, \quad \boldsymbol{\pi}^\epsilon := \underset{\boldsymbol{\pi} \in \Pi(\alpha,\beta)}{\arg\min} \langle \mathbf{D}, \boldsymbol{\pi} \rangle - \epsilon \mathrm{H}(\boldsymbol{\pi}), \quad \mathrm{H}(\boldsymbol{\pi}) := -\sum_{i,j} \boldsymbol{\pi}_{ij} \left(\log(\boldsymbol{\pi}_{ij}) - 1\right), \quad (38)$$

which makes the problem $\epsilon$-convex and solvable with the Sinkhorn algorithm (Cuturi, 2013), with a lower complexity of $\mathcal{O}(n^2/\epsilon^2)$. Besides, the Sinkhorn algorithm consists of matrix-vector products only, which makes it suited to be accelerated with GPUs. Specifically, let $\mathbf{f} \in \mathbb{R}^n$ and $\mathbf{g} \in \mathbb{R}^m$ be the lagrangian multipliers, the Lagrangian of (38) is:

$$\Phi(\boldsymbol{\pi}, \mathbf{f}, \mathbf{g}) = \langle \mathbf{D}, \boldsymbol{\pi} \rangle - \epsilon \mathrm{H}(\boldsymbol{\pi}) - \langle \mathbf{f}, \boldsymbol{\pi} \mathbf{1}_n - \mathbf{a} \rangle - \langle \mathbf{g}, \boldsymbol{\pi}^T \mathbf{1}_m - \mathbf{b} \rangle \quad (39)$$

According to the first-order condition of constraint optimization problem, we have:

$$\frac{\partial \Phi(\boldsymbol{\pi}, \mathbf{f}, \mathbf{g})}{\partial \boldsymbol{\pi}_{ij}} = \mathbf{D}_{ij} + \varepsilon \log(\boldsymbol{\pi}_{ij}) - \mathbf{f}_i - \mathbf{g}_j = 0, \quad (40)$$

or equivalently, the best transport matrix $\boldsymbol{\pi}^\epsilon$ should satisfy:

$$\boldsymbol{\pi}_{ij}^\epsilon = \exp\left(\frac{\mathbf{f}_i}{\epsilon}\right) * \exp\left(-\frac{\mathbf{D}_{ij}}{\epsilon}\right) * \exp\left(\frac{\mathbf{g}_j}{\epsilon}\right). \quad (41)$$

Let $\mathbf{u}_i := \exp(\mathbf{f}_i/\epsilon)$, $\mathbf{v}_j := \exp(\mathbf{g}_j/\epsilon)$, $\mathbf{K}_{ij} := \exp(-\mathbf{D}_{ij}/\epsilon)$, then we have $\boldsymbol{\pi}^\epsilon = \operatorname{diag}(\mathbf{u})\mathbf{K}\operatorname{diag}(\mathbf{v})$. The transport matrix should also satisfy the mass-preserving constraint, such that:

$$\operatorname{diag}(\mathbf{u})\mathbf{K}\operatorname{diag}(\mathbf{v})\mathbf{1}_m = \mathbf{a}, \qquad \operatorname{diag}(\mathbf{v})\mathbf{K}^\top\operatorname{diag}(\mathbf{u})\mathbf{1}_n = \mathbf{b}, \quad (42)$$

or equivalently, let $\odot$ be the entry-wise multiplication of vectors, we have:

$$\mathbf{u} \odot (\mathbf{Kv}) = \mathbf{a} \quad \text{and} \quad \mathbf{v} \odot (\mathbf{K}^T\mathbf{u}) = \mathbf{b}. \quad (43)$$

(43) is known as the matrix scaling problem. An intuitive approach is to solve them iteratively:

$$\mathbf{u}^{(\ell+1)} = \frac{\mathbf{a}}{\mathbf{Kv}^{(\ell)}} \quad \text{and} \quad \mathbf{v}^{(\ell+1)} = \frac{\mathbf{b}}{\mathbf{K}^T\mathbf{u}^{(\ell+1)}} \quad (44)$$

which is the critical step of Sinkhorn algorithm in Algorithm 1. The optimal transport matrix $\boldsymbol{\pi}^\epsilon$ acting as a constant matrix further induces the *Sinkhorn discrepancy* $\mathbb{W}^\epsilon$ following (38). As $\mathbf{D}$ is differentiable to $\alpha$ and $\beta$, it is feasible to minimize $\mathbb{W}^\epsilon$ by adjusting the generation process of $\alpha$ and $\beta$, *i.e.*, the representation mapping in Definition A.2 with gradient-based optimizers.

**Algorithm 2** Generalized Sinkhorn Algorithm for Unbalanced Optimal Transport

---

**Input**: discrete measures $\alpha = \sum_{i=1}^{n} \mathbf{a}_i \delta_{\mathbf{x}_i}$ and $\beta = \sum_{j=1}^{m} \mathbf{b}_j \delta_{\mathbf{x}_j}$, distance matrix $\mathbf{D}_{ij} = \|\mathbf{x}_i - \mathbf{x}_j\|_2^2$.
**Parameter**: $\epsilon$: strength of entropic regularizer; $\kappa$: strength of mass preserving; $\ell_{\max}$: max iterations.
**Output**: $\boldsymbol{\pi}^{\epsilon,\kappa}$: the entropic regularized unbalanced optimal transport matrix.

  1: $\mathbf{K} \leftarrow \exp(-\mathbf{D}/\epsilon)$.
  2: $\mathbf{f} \leftarrow \mathbf{0}_n, \mathbf{g} \leftarrow \mathbf{0}_m, \ell \leftarrow 1$.
  3: **while** $\ell < \ell_{\max}$ **do**
  4:      $\mathbf{u} \leftarrow \exp(\mathbf{f}_i/\epsilon), \mathbf{v} \leftarrow \exp(\mathbf{g}_j/\epsilon)$
  5:      $\boldsymbol{\pi} \leftarrow \operatorname{diag}(\mathbf{u})\mathbf{K}\operatorname{diag}(\mathbf{v})$.
  6:      $\mathbf{a}' \leftarrow \boldsymbol{\pi}\mathbf{1}_n, \mathbf{b}' \leftarrow \boldsymbol{\pi}^{\mathrm{T}}\mathbf{1}_m$.
  7:      **if** $\ell // 2 = 0$ **then**
  8:          $\mathbf{f} \leftarrow \left[\frac{\mathbf{f}}{\epsilon} + \log(\mathbf{a}) - \log(\mathbf{a}')\right]\frac{\epsilon\kappa}{\epsilon+\kappa}$
  9:      **else**
10:         $\mathbf{g} \leftarrow \left[\frac{\mathbf{g}}{\epsilon} + \log(\mathbf{b}) - \log(\mathbf{b}')\right]\frac{\epsilon\kappa}{\epsilon+\kappa}$
11:      $\ell \leftarrow \ell + 1$.
12: $\boldsymbol{\pi}^{\epsilon,\kappa} \leftarrow \operatorname{diag}(\mathbf{u})\mathbf{K}\operatorname{diag}(\mathbf{v})$.

---

### B.3   Unbalanced optimal transport and generalized sinkhorn

We have reported the mini-batch sampling effect (MSE) issue of $\mathbb{W}^{\epsilon}$ in Section 3.2, and attributed it to the mass-preserving constraint in (38). An intuitive approach to mitigate MSE is to relax the marginal constraint and allow for the creation and destruction of mass. To this end, RMPR is proposed in Definition B.3, which replaces the hard marginal constraint with a soft penalty.

**Definition B.3.** *For empirical distributions $\alpha$ and $\beta$ with $n$ and $m$ units, respectively, unbalanced optimal transport seeks a transport plan at minimum cost:*

$$\mathbb{W}^{\epsilon,\kappa}(\alpha,\beta) := \min_{\boldsymbol{\pi}}\langle\mathbf{D},\boldsymbol{\pi}\rangle, \boldsymbol{\pi} := \arg\min_{\boldsymbol{\pi}}\langle\mathbf{D},\boldsymbol{\pi}\rangle + \epsilon\mathrm{H}(\boldsymbol{\pi}) + \kappa(\mathbf{KL}(\boldsymbol{\pi}\mathbf{1}_n,\mathbf{a}) + \mathbf{KL}(\boldsymbol{\pi}^{\mathrm{T}}\mathbf{1}_m,\mathbf{b})), \quad (45)$$

*where $\mathbf{D} \in \mathbb{R}_{+}^{n\times m}$ is the unit-wise distance, and $\mathbf{a}$ and $\mathbf{b}$ indicate the mass of units in $\alpha$ and $\beta$.*

The unbalanced optimal transport problem in Definition B.3 has a similar structure with (38) and thus can be solved with a generalized Sinkhorn algorithm (Chizat et al., 2018). The derivation starts from the Fenchel-Legendre dual form of (45):

$$\max_{\mathbf{f}\in\mathbb{R}^n,\mathbf{g}\in\mathbb{R}^m} -F^*(-\mathbf{f}) - G^*(-\mathbf{g}) - \epsilon\sum_{i,j}\exp\left(\frac{\mathbf{f}_i + \mathbf{g}_j - \mathbf{D}_{ij}}{\epsilon}\right),$$

$$F^*(\mathbf{f}) = \max_{\mathbf{z}\in\mathbb{R}^n}\mathbf{z}^{\top}\mathbf{f} - \kappa\mathbf{KL}(\mathbf{z}\|\mathbf{a}) = \kappa\left\langle e^{\mathbf{f}/\kappa},\mathbf{a}\right\rangle - \mathbf{a}^{\top}\mathbf{1}_n, \quad (46)$$

$$G^*(\mathbf{g}) = \max_{\mathbf{z}\in\mathbb{R}^m}\mathbf{z}^{\top}\mathbf{g} - \kappa\mathbf{KL}(\mathbf{z}\|\mathbf{b}) = \kappa\left\langle e^{\mathbf{g}/\kappa},\mathbf{b}\right\rangle - \mathbf{b}^{\top}\mathbf{1}_m,$$

where the functions $F^*(\cdot)$ and $G^*(\cdot)$ are the Legendre transformation of KL divergence. Ignoring the constant terms, we can obtain the equivalent optimization problem:

$$\min_{\mathbf{f}\in\mathbb{R}^n,\mathbf{g}\in\mathbb{R}^m} \epsilon\sum_{i,j=1}^{n}\exp\left(\frac{\mathbf{f}_i + \mathbf{g}_j - \mathbf{D}_{ij}}{\epsilon}\right) + \kappa\left\langle e^{-\mathbf{f}/\kappa},\mathbf{a}\right\rangle + \kappa\left\langle e^{-\mathbf{g}/\kappa},\mathbf{b}\right\rangle. \quad (47)$$

According to the first-order condition, the minimizer's gradient of (47) should be zero. As such, fixing $\mathbf{g}^{\ell}$, the updated $\mathbf{f}^{\ell+1}$ ought to satisfy:

$$\exp\left(\frac{\mathbf{f}_i^{\ell+1}}{\epsilon}\right)\sum_{j=1}^{n}\exp\left(\frac{\mathbf{g}_j^{\ell} - \mathbf{D}_{ij}}{\epsilon}\right) = \exp\left(-\frac{\mathbf{f}_i^{\ell+1}}{\kappa}\right)\mathbf{a}_i, \quad (48)$$

We further multiply both sides by $\exp(\mathbf{f}_i^{\ell}/\epsilon)$:

$$\exp\left(\frac{\mathbf{f}_i^{\ell+1}}{\epsilon}\right)\mathbf{a}_i' = \exp\left(\frac{\mathbf{f}_i^{\ell}}{\epsilon}\right)\exp\left(-\frac{\mathbf{f}_i^{\ell+1}}{\kappa}\right)\mathbf{a}_i \quad (49)$$

---

[6]In this work, we calculate the unit-wise distance with the squared Euclidean metric following Courty et al. (2017b).

---

**Algorithm 3** ESCFR Algorithm

---

**Input**: covariates of treated units $\{\mathbf{x}_i\}_{i=1}^n$ and untreated units $\{\mathbf{x}_j\}_{j=1}^m$; factual outcomes $\{y_i\}_{i=1}^n$ and $\{y_j\}_{j=1}^m$; representation mapping $\psi$; outcome mapping $\phi$.

**Parameter**: $\lambda$: strength of optimal transport; $\epsilon$: strength of entropic regularizer; $\kappa$: strength of RMPR; $\gamma$: strength of PFOR; $\ell_{\max}$: max iterations

**Output**: $\mathcal{L}_{\mathrm{ESCFR}}^{\epsilon,\kappa,\gamma,\lambda}$: the learning objective of ESCFR.

1: $\{\mathbf{r}_i\}_{i=1}^n \leftarrow \{\psi(\mathbf{x}_i)\}_{i=1}^n, \qquad \{\mathbf{r}_j\}_{j=1}^m \leftarrow \{\psi(\mathbf{x}_j)\}_{j=1}^m.$
2: $\{\hat{y}_i\}_{i=1}^n \leftarrow \{\phi(\mathbf{r}_i,1)\}_{i=1}^n, \quad \{\hat{y}_j\}_{j=1}^m \leftarrow \{\phi(\mathbf{r}_j,0)\}_{j=1}^m.$
3: $\{\tilde{y}_i\}_{i=1}^n \leftarrow \{\phi(\mathbf{r}_i,0)\}_{i=1}^n, \quad \{\tilde{y}_j\}_{j=1}^m \leftarrow \{\phi(\mathbf{r}_j,1)\}_{j=1}^m.$
4: $\mathbf{D}_{ij}^\gamma \leftarrow \|\mathbf{x}_i - \mathbf{x}_j\|_2^2 + \gamma \cdot \|y_j - \tilde{y}_j\|_2^2 + \gamma \cdot \|y_j - \tilde{y}_i\|_2^2.$
5: $\mathbf{D}_{\mathrm{stop}}^\gamma \leftarrow \mathrm{stopgradient}(\mathbf{D}^\gamma).$
6: $\boldsymbol{\pi}^{\epsilon,\kappa,\gamma} \leftarrow \mathrm{Algorithm2}\left(\alpha = \{\mathbf{r}_i\}_{i=1}^n, \beta = \{\mathbf{r}_j\}_{j=1}^m, \mathbf{D} = \mathbf{D}_{\mathrm{stop}}^\gamma\right).$
7: $\mathcal{L}_{\mathrm{F}}(\psi,\phi) \leftarrow \frac{1}{n}\sum_{i=1}^n \|\hat{y}_i - y_i\|_2^2 + \frac{1}{m}\sum_{j=1}^m \|\hat{y}_j - y_j\|_2^2.$
8: $\mathcal{L}_{\mathrm{D}}^{\epsilon,\kappa,\gamma}(\psi) \leftarrow \langle \mathbf{D}^\gamma, \boldsymbol{\pi}^{\epsilon,\kappa,\gamma}\rangle.$
9: $\mathcal{L}_{\mathrm{ESCFR}}^{\epsilon,\kappa,\gamma,\lambda} \leftarrow \mathcal{L}_{\mathrm{F}}(\psi,\phi) + \lambda \cdot \mathcal{L}_{\mathrm{D}}^{\epsilon,\kappa,\gamma}(\psi).$

---

Table 3: Running time (mean+std) in seconds of Algorithm 1-2 with 100 runs.

| Parameter | $n = 32$ | $n = 64$ | $n = 128$ | $n = 256$ | $n = 512$ | $n = 1024$ |
|---|---|---|---|---|---|---|
| Algorithm1 | 0.0266±0.0102 | 0.0241±0.0075 | 0.0326±0.0088 | 0.0499±0.0099 | 0.0725±0.0128 | 0.1430±0.0259 |
| Algorithm2 | 0.0050±0.0004 | 0.0051±0.0001 | 0.0065±0.0002 | 0.0104±0.0005 | 0.0138±0.0008 | 0.0256±0.0007 |
| Parameter | $\epsilon = 0.1$ | $\epsilon = 0.5$ | $\epsilon = 1.0$ | $\epsilon = 5.0$ | $\epsilon = 10.0$ | $\epsilon = 100.0$ |
| Algorithm1 | 0.1683±0.0038 | 0.1207±0.0102 | 0.0699±0.0095 | 0.0153±0.0013 | 0.0097±0.0009 | 0.0072±0.0009 |
| Algorithm2 | 0.0166±0.0019 | 0.0068±0.0010 | 0.0052±0.0011 | 0.0047±0.0010 | 0.0045±0.0011 | 0.0043±0.0009 |
| Parameter | $\kappa = 0.1$ | $\kappa = 0.5$ | $\kappa = 1.0$ | $\kappa = 5.0$ | $\kappa = 10.0$ | $\kappa = 100.0$ |
| Algorithm2 | 0.0050±0.0011 | 0.0059±0.0008 | 0.0060±0.0011 | 0.0112±0.0014 | 0.0162±0.0016 | 0.1039±0.0033 |

where $\mathbf{a}' := \boldsymbol{\pi}\mathbf{1}_n$ with $\boldsymbol{\pi}_{ij} := \exp(\mathbf{f}_i^\ell + \mathbf{g}_j^\ell - \mathbf{D}_{ij})$. Similarly, fixing $\mathbf{f}$ we have $\mathbf{g}^{\ell+1}$ as:

$$\exp\left(\frac{\mathbf{g}_j^{\ell+1}}{\epsilon}\right)\mathbf{b}_j' = \exp\left(\frac{\mathbf{g}_j^\ell}{\epsilon}\right)\exp\left(-\frac{\mathbf{g}_j^{\ell+1}}{\kappa}\right)\mathbf{b}_j \tag{50}$$

where $\mathbf{b}' := \boldsymbol{\pi}^{\mathrm{T}}\mathbf{1}_m$. (49) and (50) construct the critical iteration steps of the generalized Sinkhorn algorithm (Chizat et al., 2018), which we formulate in Algorithm 2. The transport matrix $\boldsymbol{\pi}^{\epsilon,\kappa}$ further induces the *generalized Sinkhorn discrepancy* $\mathbb{W}^{\epsilon,\kappa}$ in Definition B.3. As $\mathbf{D}$ is differentiable with respect to $\alpha$ and $\beta$, it is feasible to minimize $\mathbb{W}^{\epsilon,\kappa}$ by adjusting the generation process of $\alpha$ and $\beta$, *i.e.*, the representation mapping in Definition A.2, with gradient-based optimizers.

## B.4 OPTIMIZATION OF ENTIRE SPACE COUNTERFACTUAL REGRESSION

Algorithm 3 shows how to calculate the learning objective at a mini-batch level. Specifically, we first calculate the factual outcome estimates (step 2), counterfactual outcome estimates (step 3), and the unit-wise distance matrix with PFOR (step 4). Afterwards, fix the gradient of the distance matrix (step 5) and calculate the transport matrix with Algorithm 2 (step 6). Finally, calculate the factual outcome estimation error (step 7) and distribution discrepancy (step 8), and aggregate them to acquire the learning objective of ESCFR (step 9). According to Section B.3, the learning objective is differentiable to $\psi$ and $\phi$ and thus can be optimized end-to-end with stochastic gradient methods.

## B.5 COMPLEXITY ANALYSIS

One primary concern would be the overall complexity of solving discrete optimal transport problems. Exact algorithms, *e.g.*, the interior-point method and network-simplex method, suffer from a high computational cost of $\mathcal{O}(n^3 \log n)$ (Pele & Werman, 2009). An entropic regularizer is thus introduced in (5), making the problem solvable by the Sinkhorn algorithm (Cuturi, 2013) in Algorithm 1. The

complexity was shown to be $\mathcal{O}(n^2/\epsilon^3)$ by Altschuler et al. (2017) in terms of the absolute error of the mass-preservation constraints. Dvurechensky et al. (2018) improved it to $\mathcal{O}(n^2/\epsilon^2)$, which can be further accelerated with greedy algorithm by Lin et al. (2019). Several recent explorations (Blanchet et al., 2018; Jambulapati et al., 2019) have also attempted to further reduce the complexity to $\mathcal{O}(n^2/\epsilon)$.

Entropic regularization trick is still applicable to speed up the solution of the unbalanced optimal transport problem in RMPR, represented by the Sinkhorn-like algorithm in Algorithm 2. Pham et al. (2020) further proved that the complexity of Algorithm 2 is $\tilde{\mathcal{O}}(n^2/\epsilon)$.

Table 3 reports the practical running time at the commonly-used batch settings. In general, the computational cost of optimal transmission is not a concern at the mini-batch level. Notice that enlarging $\epsilon$ speeds up the computation while making the resulting transfer matrix biased, hindering the transportation performance, as per Figure 5. In addition, a large relaxation parameter $\kappa$ makes the computed results closer to those by Sinkhorn algorithm yet significantly contributes to more iterations, which is discussed and mitigated by Séjourné et al. (2022).

## C    REPRODUCTION DETAILS

### C.1    DATASETS

We conduct experiments on two semi-synthetic benchmarks to validate our models. For the IHDP[7] benchmark, we report the results over 10 simulation realizations following Liuyi et al. (2018). However, the limited size (747 observations and 25 covariates) makes the results highly volatile. As such, we mainly validate the models on the ACIC benchmark, which was released by the ACIC-2016 competition[8]. Since the scale of the ACIC benchmark (4802 observations and 58 covariates) is much larger than the IHDP benchmark, we mainly perform ablation studies on it to get more reliable results.

All datasets are randomly shuffled and partitioned in a 0.7:0.15:0.15 ratio for training, validation, and test, where we maintain the same ratio of treated units in all three splits to avoid numerical unreliability in the validation and test phases. We find that these datasets are overly easy to fit by the model because they are semi-synthetic. To increase the distinguishability of the results, we omit preprocessing strategies, such as min-max scaling, to increase the difficulty of the learning task.

### C.2    BASELINES

We compare the proposed method with baselines based on statistical estimators (Künzel et al., 2019; Shalit et al., 2017), matching estimators (Rosenbaum & Rubin, 1983b; Wager & Athey, 2018; Crump et al., 2008) and representation-based estimators (Johansson et al., 2016; Shalit et al., 2017). We implement all these baselines by hand, largely built upon Pytorch for neural network models, Sklearn for statistical models, and EconML for tree and forest models. These implementations will be open-sourced.

### C.3    METRICS

Following existing works (Yao et al., 2019; Liuyi et al., 2018), the Precision in Estimation of Heterogeneous Effect (PEHE) is primarily used as the precision metric for performance evaluation.

$$\epsilon_{\text{PEHE}}(\psi, \phi) = \int_{\mathcal{X}} \left( \hat{\tau}_{\psi,\phi}(x) - \tau(x) \right)^2 \mathbb{P}(x)\, dx. \tag{51}$$

However, PEHE is unavailable during the model selection phase as the counterfactual outcomes are missing. A shortcut adapted by Liuyi et al. (2018) for model selection would be the root mean squared error of factual outcome estimates ($\text{RMSE}_{\text{F}}$), which can be evaluated in the absence of the counterfactual outcomes. However, owing to the treatment selection bias, $\text{RMSE}_{\text{F}}$ is not reliable as it does not consider the precision of counterfactual estimation, and thus cannot effectively evaluate the quality of treatment estimation.

---

[7]It can be downloaded from https://www.fredjo.com/

[8]It can be downloaded from https://jenniferhill7.wixsite.com/acic-2016/competition

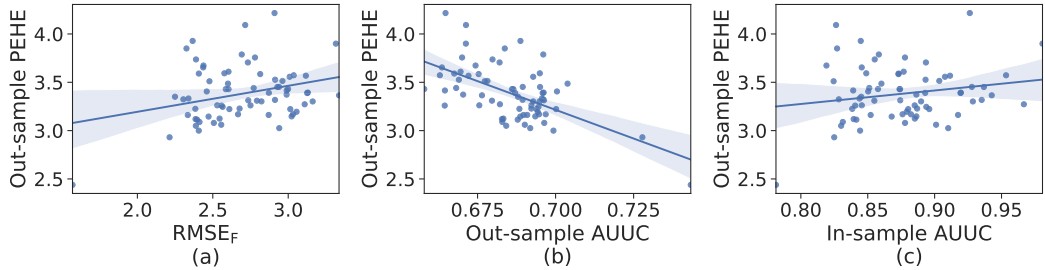

Figure 7: Fidelity of model-selection criterion to out-of-sample PEHE

Area Under the Uplift Curve (AUUC) (Betlei et al., 2021) evaluates the model's performance in ranking units based on potential treatment benefit. It is more feasible than PEHE because it can be calculated without counterfactual outcomes; it is more reliable than factual MSE because it partially reflects the models' counterfactual ranking ability. Therefore, it has been the primary selection standard by practitioners from Criteo (Betlei et al., 2021), Alibaba (Ke et al., 2021), and Tencent (He et al., 2022), where engineers always fine-tune their causal inference models concerning AUUC performance and decide whether to deploy the models to online traffic. As such, we use AUUC as the model selection criteria. Moreover, we report the within-sample results on the training dataset and the out-of-sample results on the test dataset, where the factual outcome is available in the within-sample case following Shalit et al. (2017).

We compare the fidelity of $\mathrm{RMSE_F}$ and AUUC on the evaluation set to the out-of-sample PEHE on the test set in Figure 7. The first observation is the weak correlation between $\mathrm{RMSE_F}$ on the validation set and the out-of-sample PEHE on the test set. In contrast, the out-of-sample AUUC on the evaluation set could better reflect the variation of PEHE. Another observation is that the within-sample AUUC is not a good criterion for model selection, as higher within-sample AUUC corresponds to higher out-of-sample PEHE, which contrasts with reality. This is reasonable as better within-sample performance does not necessarily correspond to better out-of-sample performance.

## D    ADDITIONAL DISCUSSIONS

### D.1    ADDITIONAL DISCUSSION FOR STOCHASTIC OPTIMAL TRANSPORT

According to Theorem 3.1, one critical hyperparameter for CFR-WASS and ESCFR is the batch size, which directly affects the variance of stochastic optimal transport in Section 3.1 and thus the performance of both methods. As such, it is necessary to verify whether ESCFR outperforms CFR for different batch sizes. We conduct extensive experiments and summarize the results in Table 4. Interesting observations are noted:

- Increasing batch size in a wide range improves the performance of CFR-WASS and ESCFR. For example, The PEHE of CFR-WASS decreases from 3.114 at $b = 32$ to 2.932 at $b = 128$, and the PEHE of ESCFR exhibits a similar pattern. The performance gain is attributed to the decreased variance in (6), which backs up Theorem 3.1.

- By finetuning batch size, we can easily exceed the performance we report in Table 1. However, we did not finetune it as the PEHE is invisible during our hyper-parameter tuning process[9].

- The performance drop for the overly large batch sizes comes from the sub-optimal backbone (TARNet) performance. Due to the limited number of training samples, *e.g.*, 4.8k * 70% units for ACIC and 0.7k * 70% units for IHDP, a large batch size might block the models from escaping saddle points (Jin et al., 2017) and sharp minima (Xie et al., 2020), and thus deteriorating the accuracy of factual outcome estimation.

---

[9]Most of the experiments in Table 1 were performed with a fixed batch size $b = 32$, which is selected by the factual estimation performance of TARNet.

Table 4: Out-of-sample PEHE of ESCFR and important baselines with different batch sizes $b$.

| Model | $b = 32$ | $b = 64$ | $b = 96$ | $b = 128$ | $b = 196$ | $b = 256$ |
|---|---|---|---|---|---|---|
| TARNet | 3.3293±0.1853 | 3.2054±0.2676 | 3.0869±0.2812 | 2.9262±0.3160 | 3.4619±0.6652 | 3.6309±0.2026 |
| CFR-WASS | 3.1143±0.4578 | 3.0819±0.3407 | 2.9998±0.1017 | 2.9326±0.4142 | 4.0740±1.4127 | 3.4675±0.1552 |
| ESCFR | 2.3736±0.1621 | 2.3082±0.4334 | 2.9719±0.2889 | 2.3125±0.1836 | 2.0373±0.1538 | 2.2777±0.4230 |

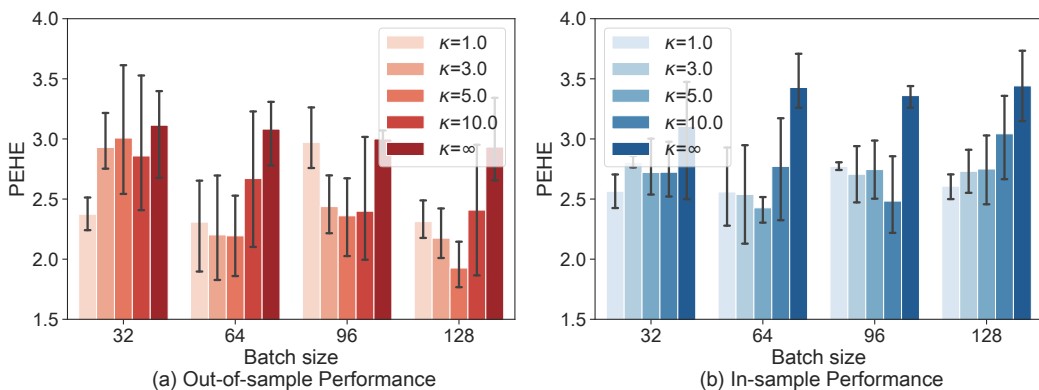

Figure 8: PEHE of ESCFR and CFR-WASS ($\kappa = \infty$) under different batch size.

## D.2 ADDITIONAL DISCUSSION FOR RELAXED MASS-PRESERVING REGULARIZER

Existing methods (Shalit et al., 2017; Johansson et al., 2016; Liuyi et al., 2018) suffer from the mini-batch sampling effect (MSE) issue, as indicated by the two bad cases in Figure 2. RMPR mitigates the MSE issue by relaxing the mass-preserving constraint, the performance of which is affected by two critical hyperparameters, *i.e.*, the batch size $b$ and the strength of mass-preserving constraint $\kappa$. On top of the ablation studies, it is necessary to explore the performance of ESCFR at different settings of $b$ and $\kappa$, to investigate 1) how RMPR works; 2) the limitation and bottleneck of RMPR; 3) the robustness of RMPR to hyperparameter setting. The results are presented in Figure 8, and the observations are summarized as follows.

- The optimal value of $\kappa$ increases with the increase of batch size. For example, the optimal $\kappa$ is 1.0 at $b = 32$, and 5.0 at $b = 128$. This observation partially verifies how RMPR works as described in Section 3.2. Specifically, at small batch sizes where sampling outliers dominate the sampled batches, a small $\kappa$ effectively relaxes the mass-preserving constraint and avoids the damage of mini-batch outliers, thus improving the performance effectively and robustly. At large batch sizes, the noise of sampling outliers is reduced, and it is reasonable to increase $\kappa$ to match more units and obtain more accurate wasserstein distance estimates.

- Even with large batch sizes, oversized $\kappa$, *e.g.*, $\kappa \geq 10$ does not perform well. Although the effect of sampling outliers is reduced, some patterns such as outcome imbalance are present for all batch sizes, resulting in false matching given large mass-preserving constraint strength $\kappa$, which might be a primary bottleneck of RMPR.

- Hyper-parameter tuning is not necessarily the reason why ESCFR works well, since all ESCFR implementations outperform the strongest baseline CFR-WASS ( $\kappa = \infty$) on all batch sizes, most of which are statistically significant. This can be further supported by our extensive ablation study in Section 4.3 and parameter study in Section 4.5.

In summary, it is necessary to relax the mass-preserving constraint under all settings of batch size, which strongly verifies the effectiveness of RMPR in Section 3.2.

## D.3 ADDITIONAL DISCUSSION FOR PROXIMAL FACTUAL OUTCOME REGULARIZER

Existing representation-based methods block the backdoor path $X \rightarrow T$ by balancing the distribution of the observed covariates in a latent space. Given the unconfoundedness assumption A.1, this

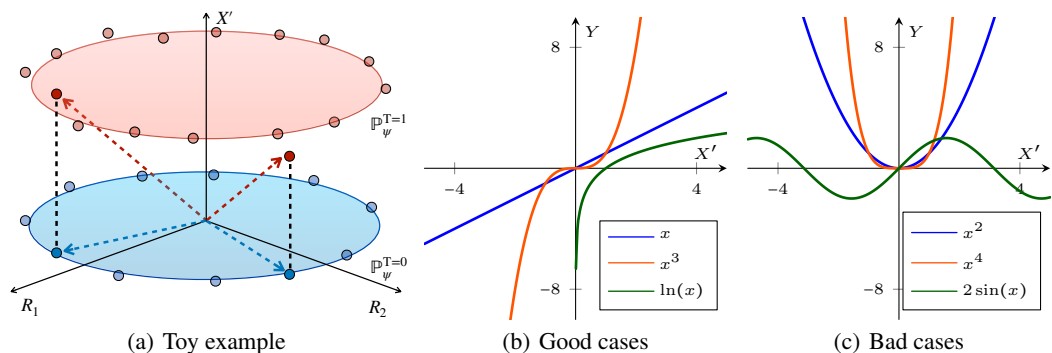

| (a) Toy example | (b) Good cases | (c) Bad cases |

Figure 9: A diagram showing how PFOR works and its limitations. (a) A toy example of PFOR, where $R$ and $X'$ indicate the balanced representations and an unobserved confounder, respectively; scatters indicate the empirical distribution of units in the treated and control groups; for solid scatters with balanced $R$, the colored dashed line indicates the ground truth outcome $Y = \sqrt{R_1^2 + R_2^2 + X'^2}$ in each group, the black dashed line measures the difference of unobserved $X'$. (b) Cases that satisfy Assumption D.1, where the the outcome $Y$ is monotone with unobserved $X'$ given observed confounders in $R$. (c) Cases that violate Assumption D.1, where the $Y$ is non-monotone with $X'$.

approach effectively handles the treatment selection bias. However, Assumption A.1 is usually violated in practice, which invalidates this approach as the backdoor path from the unobserved confounder $X'$ to $T$ is not blocked.

According to the designed causal graph in Figure 3(b), all factors associated with outcome $Y$ include the observed confounders $X$, treatment $T$, and unobserved confounders $X'$. Therefore, it is reasonable to derive that given balanced $X$ and identical $T$, the only variable reflecting the variation of $X'$ is the outcome $Y$. As such, inspired by the joint distribution transport technique (see Courty et al., 2017a), PFOR calibrates the unit-wise distance $\mathbf{D}$ with the potential outcomes in (12). The underlying regularization is: units with similar (observed and unobserved) confounders should have similar potential outcomes. Equivalently, for a pair of units with similar observed covariates, *i.e.*, $\|r_i - r_j\|^2 \approx 0$, if their potential outcomes under the same treatment $t = \{0, 1\}$ differ significantly, *i.e.*, $\|y_i^t - y_j^t\| \gg 0$, their unobserved confounders should also differ significantly. As such, it is reasonable to utilize the difference of outcomes to calibrate the unobserved confounding effect.

**Assumption D.1.** *(Monotonicity). For all observed covariates $X = x$ in the population of interest, let $T = t$ and $X' = x'$ be the treatment assignment and unobserved confounders, respectively, we have $\mathbb{E}[Y \mid X = x, X' = x', T = t]$ is monotonically increasing or decreasing with respect to $x'$.*

**Advantages.** The advantages of PFOR can be further interpreted as follows.

- From a statistical perspective, PFOR encourages units with similar outcomes to share similar representations. It is a valid prior that inspires many learning algorithms, *e.g.*, K-nearest neighbors and gaussian process (see Williams & Rasmussen, 2006). As an effective statistical regularizer, PFOR also works in the absence of unobserved confounders, especially on small data sets.

- From a domain adaptation perspective, vanilla Sinkhorn aligns the distributions $\mathbb{P}_\psi^{T=1}(r)$ and $\mathbb{P}_\psi^{T=0}(r)$, where $r$ is the learned representations in Definition A.4. PFOR further aligns the transition probabilities $\mathbb{P}^{T=1}(Y(T = t) \mid r)$ and $\mathbb{P}^{T=0}(Y(T = t) \mid r)$ for $t = 0, 1$. The discrepancy between transition probabilities can be attributed to unobserved confounders that can be viewed as parameters of the transition probabilities (Courty et al., 2017a). As such, it is feasible to align the unobserved confounders by aligning the transition probabilities.

**Toy example.** Let the ground truth $Y := \sqrt{R_1^2 + R_2^2 + X'^2}$ where $T$ is omitted as we only consider one group, $R_1$ and $R_2$ are the representations of observed confounders that have been aligned with Sinkhorn algorithm. Let the unobserved $X' = 0$ for controlled units and $X' = 1$ for treated units,

Table 5: RMSE of ESCFR and its competitors for factual and counterfactual outcome estimation.

| Dataset | Outcome | TARNet | CFR-WASS | ESCFR |
|---------|---------|--------|----------|-------|
| Train | Factual | 2.2066±0.1340 | 2.1744±0.2179 | 2.1608±0.2114 |
| Train | Counterfactual | 3.4731±0.1493 | 3.4306±0.0904 | 2.7025±0.1109 |
| Test | Factual | 3.1924±0.0357 | 3.1887±0.1493 | 2.6334±0.2253 |
| Test | Counterfactual | 3.5577±0.0957 | 3.4163±0.0496 | 2.7976±0.1378 |

which makes $X'$ an unobserved confounder as it is related to Y and different between groups. As shown in Figure 9(a), given balanced $R_1$ and $R_2$, the variation of $Y$ reveals that of $X'$. As such, it is reasonable to employ $Y$ to calibrate the unit-wise distance $\mathbf{D}$ that ignores $X'$.

**Synthetic labels.** PFOR remains effective for semi-synthetic data, where outcomes are synthetic from the covariates and treatment assignments. One source of hidden confounders in such data is the information loss from the raw data space to the representation space, where not all valuable information (*e.g.*, confounders) is extracted and preserved. Besides, this improvement could also come from the statistical regularization, encouraging units with similar outcomes to share similar representations, which is an effective prior according to the K-nearest neighboring methods.

**Limitations.** PFOR fails for confounders adding constant effects to all units. Specifically, for unobserved confounder $X'$ and treatment assignment $t = 0, 1$, if $\mathbb{E}[Y \mid X, X' = x_1, T = t] = \mathbb{E}[Y \mid X, X' = x_2, T = t]$, PFOR fails to eliminate the confounding effect of $X'$. Examples can be found in Figure 9 (c). Nevertheless, in real scenarios, it is rare that different values of $X'$ only add a constant effect to the outcome (see Sofer et al., 2016; Zheng et al., 2021; Ogburn & VanderWeele, 2012), making PFOR still effective in a wide range of application scenarios.

This limitation is formalized as Assumption D.1, where the outcome should be monotonically increasing or decreasing with unobserved confounders given observed confounders and treatment assignment, as shown in Figure 9 (b). Notably, it is a commonly used assumption in confounder analysis (Sofer et al., 2016; Zheng et al., 2021). Besides, this assumption is often plausible, at least approximately, conditional on $T = t$ (Zheng et al., 2021) . For example, it naturally holds for binary confounders; and generally holds in applications such as epidemiology (Ogburn & VanderWeele, 2012). Finally, this assumption is only imposed on the hidden confounder $X'$ following Zheng et al. (2021), which further weakens Assumption D.1 significantly.

## D.4  ADDITIONAL DISCUSSION FOR PERFORMANCE IMPROVEMENT

In practice, ESCFR serves as an efficient regularizer on TARNet. To further investigate its regularization effects, the RMSE for estimating factual and counterfactual outcomes on the training and test sets is reported in Table 5. We find the effectiveness of ESCFR comes from the three sources:

- Better fitting the training set, where ESCFR achieves the minimum RMSE for factual outcomes on the training set. This improvement is attributed to RMPR, which mitigates the mini-batch sampling effect, thus reducing mismatching.
- Effective statistical regularization, where ESCFR reduces the gap between the RMSE for factual outcomes on the training set and that on the test set by 52.1% compared with TARNet. It is mainly attributed to PFOR, which encourages samples with similar outcomes to share similar representations.
- Effective counterfactual regularization, where ESCFR reduces the gap between the RMSE on the test set for factual outcomes and that for counterfactual outcomes by 55.1% based on TARNet; the gap between the RMSE on the training set for factual outcomes and that for counterfactual outcomes by 57.3%. It makes ESCFR robust to the treatment selection bias.

## D.5  ADDITIONAL COMPARISON RESULTS

In addition to the competitors in Table 1, we add two cutting-edge representation-based ITE estimators to bring our scope of baselines up to date. Instead of designing better alignment methods, the additional baselines introduce variable decomposition and employ the distribution alignment

Table 6: Additional comparison (mean±std) on the ACIC benchmark.

| Dataset | Out-sample | | In-sample | |
| --- | --- | --- | --- | --- |
| Model | PEHE | AUUC | PEHE | AUUC |
| TARNet | 3.2054±0.2676 | 0.6683±0.0139 | 3.2933±0.4076 | **0.8597±0.0350** |
| CFR-WASS | 3.0819±0.3407 | 0.6693±0.0250 | 3.4280±0.2857 | 0.8439±0.0349 |
| DRCFR | 3.0199±0.0658 | 0.6360±0.0080 | 3.4957±0.0895 | 0.8456±0.0073 |
| DRCFR-MIM | 2.9593±0.1532 | 0.6456±0.0189 | 3.3875±0.2208 | 0.8434±0.0093 |
| ESCFR | **2.3082±0.4334** | **0.7311±0.0123** | **2.5385±0.5020** | 0.8416±0.0367 |

technology (CFR) as a component. Therefore, *it would be unfair for ESCFR to compare with them, as ESCFR is not equipped with the variable decomposition technology*. As a result, these results are not included in Table 1.

**Baselines.**    The first additional baseline is DRCFR (Hassanpour & Greiner, 2020) . It first decomposes the latent space into instrument variables, confounding variables, and adjustment variables, and then uses MMD distance to align the distribution of the adjustment variables. The second additional baseline is the MIM-DRCFR (Cheng et al., 2022). As the SOTA approach as recent as 2022, it augments DRCFR with three orthogonal constraints and uses the Wasserstein distance to align the distribution of adjustment variables. We highlight that neither of the two methods considers how to mitigate the mini-batch sampling effect and unobserved confounder effect, the two key problems in representation-based causal inference. Therefore, their contributions are quite different from our study.

**Settings.**    Since the full implementations of both baselines are not released, we reproduce them largely from scratch. Nevertheless, DRCFR released its model structure based on tensorflow 1.x, which gives us an important guideline. Experiments are conducted on the ACIC benchmark. In terms of parameters shared with CFR, we set them to the values of CFR and ESCFR for fairness. These shared parameters mainly include the learning rate, weight decay, batch size, strength of distribution alignment, etc. Notably, we set the batch size to 64 instead of 32 in Table 1, to highlight the advantages of these representation-based baselines. In terms of model structure, the Shared-bottom and Factor-specific Disentangled representation module consists of a shared fully connected layer (60 units) cascaded by three parallel fully connected layers (30 units per layer). The outcome prediction modules consist of two fully connected layesr (60 units per layer) following TARNet, CFR, and ESCFR. Empirically, it is a good setup with competitive performance and a similar number of learnable parameters as ESCFR, ensuring a fair comparison.

**Results.**    As shown in Table 6, the additional baselines show a more stable performance (less variance) and do outperform CFR-WASS in terms of out-of-sample PEHE. However, ESCFR outperforms the additional baselines significantly in terms of most metrics. The only exception, in-sample AUUC, has been analyzed in Section 4.

## D.6    ADDITIONAL RESULTS FOR ABLATION STUDY

In this section, we provide additional results on hyperparameter study [10] as it reflects how different components of ESCFR affect the performance. Specifically, we report the ranking performance under different settings in Figure 10 and Table 7. The out-of-sample AUUC exhibits converse variation patterns with PEHE, which is consistent with our expectations. We add the estimation error of average treatment effect estimation for the entire population ($\varepsilon_{\mathrm{ATE}}$) and treated population ($\varepsilon_{\mathrm{ATT}}$) in Figure 11, Figure 12 and Table 8, where the PFOR significantly reduces both average estimation errors. Finally, we report the root mean squared error of factual outcomes ($\mathrm{RMSE_F}$) and counterfactual outcomes ($\mathrm{RMSE_F}$). Overall, all components of ESCFR could reduce $\mathrm{RMSE_{CF}}$ significantly. One interesting observation is that the dropping of $\mathrm{RMSE_{CF}}$ always comes with the dropping of $\mathrm{RMSE_F}$. It shows that distribution adjustment does not sacrifice the performance of the factual outcome estimators, but provides prompting information that improves both factual and counterfactual outcome estimators.

---

[10]We take a 40% confidence interval to plot error bars to highlight trends. Exact values are given in tables.

Table 7: Individual treatment effect estimation performance (mean±std) of ESCFR under different settings.

| Metric | PEHE | | AUUC | |
|---|---|---|---|---|
| Model | In-sample | Out-sample | In-sample | Out-sample |
| $\lambda = 0.0$ | 3.2367±0.2666 | 3.2542±0.1505 | 0.8862±0.0462 | 0.6624±0.0149 |
| $\lambda = 0.5$ | 2.8069±0.3913 | 2.9891±0.4981 | 0.8309±0.0298 | 0.6965±0.0166 |
| $\lambda = 1.0$ | 2.7324±0.2716 | 2.9053±0.3744 | 0.8312±0.0262 | 0.7022±0.0223 |
| $\lambda = 1.5$ | 2.6522±0.2605 | 2.8434±0.5605 | 0.8222±0.0385 | 0.7114±0.0187 |
| $\lambda = 2.0$ | 2.7580±0.3065 | 2.8741±0.4639 | 0.8427±0.0412 | 0.6978±0.0181 |
| $\lambda = 3.0$ | 2.6771±0.2088 | 2.8807±0.4148 | 0.8305±0.0317 | 0.7088±0.0201 |
| $\epsilon = 0.5$ | 2.6855±0.2579 | 2.8640±0.4057 | 0.8285±0.0291 | 0.7039±0.0199 |
| $\epsilon = 1.0$ | 2.6842±0.3529 | 2.9532±0.5444 | 0.8383±0.0223 | 0.7042±0.0248 |
| $\epsilon = 1.5$ | 2.7913±0.2336 | 3.0501±0.4512 | 0.8543±0.0358 | 0.7042±0.0192 |
| $\epsilon = 2.0$ | 2.7324±0.2716 | 2.9053±0.3744 | 0.8312±0.0262 | 0.7022±0.0223 |
| $\epsilon = 2.5$ | 2.8486±0.2229 | 3.1263±0.3486 | 0.8521±0.0263 | 0.6945±0.0172 |
| $\epsilon = 3.0$ | 2.7025±0.3612 | 2.9353±0.4180 | 0.8417±0.0470 | 0.6946±0.0162 |
| $\kappa = 5$ | 2.6695±0.2931 | 2.8612±0.5707 | 0.8395±0.0315 | 0.7096±0.0193 |
| $\kappa = 10$ | 2.6598±0.2627 | 2.7782±0.3791 | 0.8396±0.0299 | 0.7085±0.0155 |
| $\kappa = 15$ | 2.6024±0.2526 | 2.7582±0.5217 | 0.8276±0.0219 | 0.7065±0.0196 |
| $\kappa = 20$ | 2.6140±0.2392 | 2.7201±0.4249 | 0.8362±0.0279 | 0.7059±0.0191 |
| $\kappa = 25$ | 2.8598±0.3253 | 2.9556±0.3420 | 0.8451±0.0318 | 0.6990±0.0167 |
| $\gamma = 0.0$ | 2.7324±0.2716 | 2.9053±0.3744 | 0.8312±0.0262 | 0.7022±0.0223 |
| $\gamma = 0.1$ | 2.6120±0.3542 | 2.8305±0.5846 | 0.8272±0.0392 | 0.7064±0.0223 |
| $\gamma = 0.5$ | 2.6084±0.3425 | 2.6779±0.4628 | 0.8267±0.0320 | 0.7095±0.0275 |
| $\gamma = 1.0$ | 2.6067±0.3012 | 2.7940±0.4347 | 0.8317±0.0501 | 0.7107±0.0204 |
| $\gamma = 5.0$ | 2.6920±0.3115 | 2.7865±0.4249 | 0.8334±0.0276 | 0.7081±0.0234 |
| $\gamma = 10.0$ | 2.7643±0.3658 | 2.9006±0.5537 | 0.8494±0.0556 | 0.7037±0.0269 |

Table 8: Average treatment estimation error (mean±std) of ESCFR under different settings.

| Metric | $\varepsilon_{\mathrm{ATE}}$ | | $\varepsilon_{\mathrm{ATT}}$ | |
|---|---|---|---|---|
| Model | In-sample | Out-sample | In-sample | Out-sample |
| $\lambda = 0.0$ | 0.2930±0.3176 | 0.8431±0.3681 | 0.6688±0.6645 | 0.6880±0.3808 |
| $\lambda = 0.5$ | 0.3108±0.2974 | 0.8235±0.7135 | 0.4830±0.4356 | 0.7677±0.6753 |
| $\lambda = 1.0$ | 0.3779±0.2954 | 0.8992±0.6360 | 0.5760±0.4385 | 0.7802±0.6336 |
| $\lambda = 1.5$ | 0.5184±0.2416 | 1.0470±0.7141 | 0.7806±0.4253 | 0.9899±0.6105 |
| $\lambda = 2.0$ | 0.2898±0.2162 | 0.9060±0.6212 | 0.4325±0.2883 | 0.7868±0.5812 |
| $\lambda = 3.0$ | 0.4088±0.3408 | 1.1433±0.6801 | 0.5424±0.5269 | 1.0604±0.6303 |
| $\epsilon = 0.5$ | 0.2561±0.2307 | 0.8395±0.5324 | 0.3989±0.3687 | 0.7391±0.4853 |
| $\epsilon = 1.0$ | 0.2617±0.1962 | 1.0247±0.4543 | 0.3512±0.2772 | 0.9203±0.4515 |
| $\epsilon = 1.5$ | 0.3353±0.2556 | 1.2782±0.5472 | 0.4339±0.3214 | 1.1656±0.5019 |
| $\epsilon = 2.0$ | 0.3779±0.2954 | 0.8992±0.6360 | 0.5760±0.4385 | 0.7802±0.6336 |
| $\epsilon = 2.5$ | 0.4934±0.2857 | 1.3844±0.5709 | 0.4625±0.4451 | 1.2784±0.5561 |
| $\epsilon = 3.0$ | 0.2889±0.1677 | 1.0642±0.5469 | 0.3765±0.2717 | 0.9154±0.5379 |
| $\kappa = 5$ | 0.3598±0.3597 | 1.0892±0.7782 | 0.3969±0.3444 | 1.0047±0.7279 |
| $\kappa = 10$ | 0.2964±0.2380 | 0.8721±0.6014 | 0.4106±0.2212 | 0.7972±0.5751 |
| $\kappa = 15$ | 0.3797±0.2402 | 0.9187±0.6839 | 0.5244±0.3445 | 0.8463±0.6493 |
| $\kappa = 20$ | 0.2799±0.2918 | 0.8306±0.5746 | 0.3740±0.4330 | 0.7502±0.5503 |
| $\kappa = 25$ | 0.2469±0.1943 | 0.6571±0.5349 | 0.5481±0.4195 | 0.6031±0.4486 |
| $\gamma = 0.0$ | 0.3779±0.2954 | 0.8992±0.6360 | 0.5760±0.4385 | 0.7802±0.6336 |
| $\gamma = 0.1$ | 0.3176±0.3216 | 0.7459±0.7342 | 0.5524±0.3777 | 0.6759±0.6982 |
| $\gamma = 0.5$ | 0.2170±0.1470 | 0.6312±0.4785 | 0.3740±0.3230 | 0.5505±0.4044 |
| $\gamma = 1.0$ | 0.2371±0.1785 | 0.8427±0.6201 | 0.3036±0.2446 | 0.7764±0.5583 |
| $\gamma = 5.0$ | 0.3903±0.3032 | 0.8365±0.6857 | 0.6347±0.5429 | 0.7601±0.6364 |
| $\gamma = 10.0$ | 0.2977±0.1908 | 0.8416±0.4937 | 0.5226±0.4002 | 0.7335±0.4856 |

Table 9: Outcome estimation percision (mean±std) of ESCFR under different settings.

| Metric | RMSE$_F$ | | RMSE$_{CF}$ | |
|---|---|---|---|---|
| Model | In-sample | Out-sample | In-sample | Out-sample |
| $\lambda = 0.0$ | 2.7785±0.3642 | 3.2952±0.2815 | 3.2367±0.2666 | 3.4277±0.3103 |
| $\lambda = 0.5$ | 2.2107±0.4151 | 2.6970±0.3937 | 2.8069±0.3913 | 2.9530±0.4813 |
| $\lambda = 1.0$ | 2.1674±0.3397 | 2.6953±0.3274 | 2.7324±0.2716 | 2.8730±0.2980 |
| $\lambda = 1.5$ | 2.1288±0.3921 | 2.6346±0.3601 | 2.6522±0.2605 | 2.8538±0.3549 |
| $\lambda = 2.0$ | 2.3265±0.4361 | 2.7961±0.4254 | 2.7580±0.3065 | 2.9292±0.3954 |
| $\lambda = 3.0$ | 2.2182±0.3445 | 2.6948±0.2941 | 2.6771±0.2088 | 2.8553±0.2897 |
| $\epsilon = 0.5$ | 2.1481±0.3961 | 2.6331±0.3469 | 2.6855±0.2579 | 2.8517±0.3782 |
| $\epsilon = 1.0$ | 2.1540±0.3650 | 2.6842±0.3800 | 2.6842±0.3529 | 2.8743±0.4060 |
| $\epsilon = 1.5$ | 2.3866±0.4056 | 2.8419±0.3468 | 2.7913±0.2336 | 2.9439±0.3116 |
| $\epsilon = 2.0$ | 2.1674±0.3397 | 2.6953±0.3274 | 2.7324±0.2716 | 2.8730±0.2980 |
| $\epsilon = 2.5$ | 2.4194±0.3648 | 2.8643±0.2888 | 2.8486±0.2229 | 2.9970±0.2555 |
| $\epsilon = 3.0$ | 2.2324±0.5539 | 2.7804±0.4149 | 2.7025±0.3612 | 2.8609±0.3312 |
| $\kappa = 5$ | 2.2200±0.5065 | 2.6801±0.4200 | 2.6695±0.2931 | 2.8626±0.3787 |
| $\kappa = 10$ | 2.1754±0.3667 | 2.6486±0.2866 | 2.6598±0.2627 | 2.8025±0.3281 |
| $\kappa = 15$ | 2.0850±0.3475 | 2.5694±0.3441 | 2.6024±0.2526 | 2.7893±0.2970 |
| $\kappa = 20$ | 2.0867±0.3410 | 2.5790±0.2977 | 2.6140±0.2392 | 2.7546±0.3400 |
| $\kappa = 25$ | 2.3251±0.3592 | 2.7818±0.3168 | 2.8598±0.3253 | 3.0228±0.2797 |
| $\gamma = 0.0$ | 2.1674±0.3397 | 2.6953±0.3274 | 2.7324±0.2716 | 2.8730±0.2980 |
| $\gamma = 0.1$ | 2.0466±0.5003 | 2.5924±0.4407 | 2.6120±0.3542 | 2.8192±0.4226 |
| $\gamma = 0.5$ | 2.0507±0.4153 | 2.5509±0.4264 | 2.6084±0.3425 | 2.7283±0.4048 |
| $\gamma = 1.0$ | 2.0888±0.5184 | 2.5788±0.4708 | 2.6067±0.3012 | 2.7722±0.3143 |
| $\gamma = 5.0$ | 2.1111±0.3639 | 2.6112±0.3557 | 2.6920±0.3115 | 2.7858±0.3228 |
| $\gamma = 10.0$ | 2.1696±0.4620 | 2.6797±0.4161 | 2.7643±0.3658 | 2.8965±0.5432 |

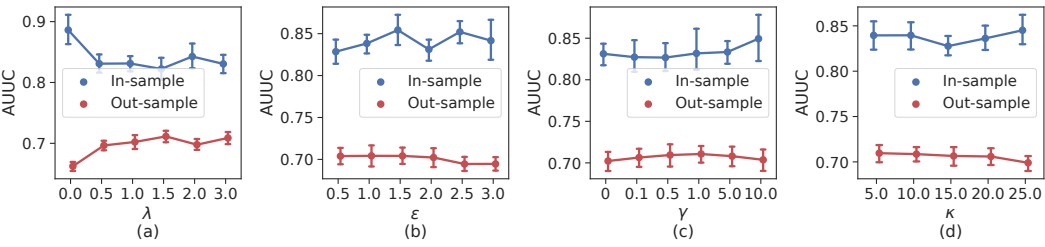

Figure 10: Parameter sensitivity study for critical hyper-parameters of ESCFR (AUUC).

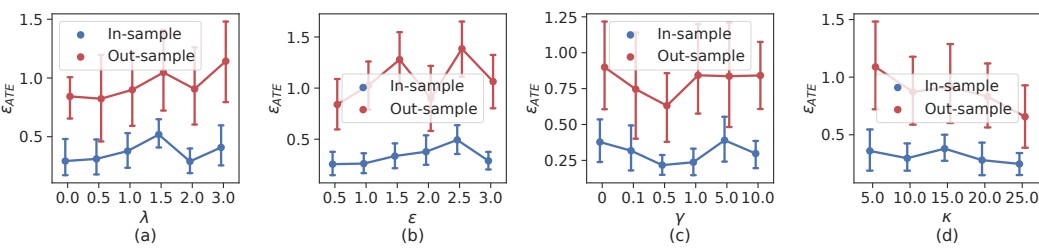

Figure 11: Parameter sensitivity study for critical hyper-parameters of ESCFR ($\varepsilon_{\text{ATE}}$).

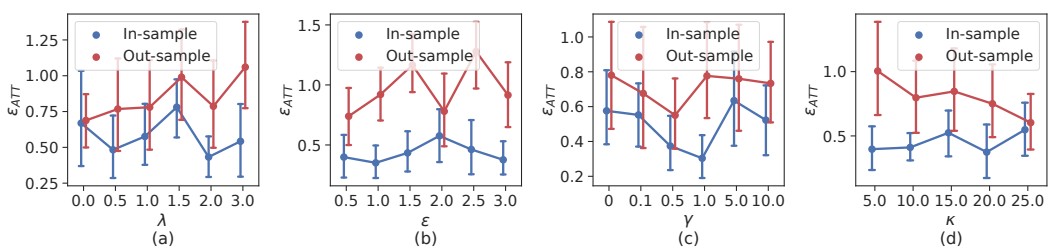

Figure 12: Parameter sensitivity study for critical hyper-parameters of ESCFR ($\varepsilon_{\text{ATT}}$).

### D.7 ADDITIONAL DISCUSSION FOR THE BASIC ASSUMPTIONS IN CAUSALITY

This section reviews the fundamental assumptions in causality and illustrates their relationship to ESCFR.

- The positivity assumption A.3 implies that the treated and untreated groups should contain overlapping units. The stochastic optimal transport in Section 3.1 seeks to achieve it in the latent representation space; however, the MSE issue leads to outcome imbalance and sampled outliers at a mini-batch level. In this case, pulling all samples into a common overlapping region leads to incorrect matches and thus misleads the update of the representation mapping $\psi$. The RMPR adaptively matches and aligns the units that are close to the overlapping region by ignoring outliers, which reduces incorrect matches and prevents biased update of $\psi$.

- The PFOR is closely related to the unconfoundedness assumption A.1. On the basis of the vanilla Sinkhorn in Section 3.1, it further aligns the transition probabilities $\mathbb{P}^{T=1}(Y(T=t) \mid r)$ and $\mathbb{P}^{T=0}(Y(T=t) \mid r)$. That is, for a pair of units from different treatment groups, if they share similar confounders $r$, their potential outcomes, *i.e.*, $Y^0, Y^1$, should also be similar. In other words, the potential outcomes are independent of the specific treatment assignment given confounders $r$ obtained with the assistance of PFOR, *i.e.*, $(Y^0, Y^1) \perp\!\!\!\perp T \mid r$, which is exactly the unconfoundedness assumption A.1.