# OpenReview forum: "Sinkhorn Discrepancy for Counterfactual Generalization"
_ICLR.cc/2023/Conference — Submitted to ICLR 2023_

### Official Review · Reviewer_VvUp · 2022-10-23

**Confidence:** 3
**Correctness:** 3
**Technical Novelty And Significance:** 2
**Empirical Novelty And Significance:** 2
**Recommendation:** 5

**Clarity, Quality, Novelty And Reproducibility:**

The paper isn't very clear and it took me a lot of effort to understand it. For example, how do you align r_i and r_j in Section 3.3? Is the algorithm affected by this at all?

The PFOR is a nice algorithm that should be interesting to study theoretically. Can the authors clarify how novel the algorithm is?

The code is not published in the supplementary and I didn't find a link to it - so Id give it a lower score for reproducibility

**Strength And Weaknesses:**

Strengths:

- PFOR is a nice technique to mitigate the missing confounder problem
- Performance is good on test sets

Weaknesses:

- I think PFOR is the most interesting part of this paper. A lot of counterfactual analysis comes down to the issue of missing confounders. It seems assuming a monotonic effect is all PFOR needs but I didn't see any theorem proving this

**Summary Of The Paper:**

In this work the authors propose an optimal transport framework algorithm called Entire Space Counter-Factual Regression (ESCFR). The method is interesting in itself, with the additions the authors propose (specially the PFOR) add some light on how neural representations could be better used for counterfactual estimation

**Summary Of The Review:**

Overall, this paper presents a nice idea of how to use neural representations for counterfactual estimation. I really enjoyed the PFOR idea and thats part of why I'm leaning towards an accept

I think more empirical studies are merited - and Id like to see a proof of what is the minimal set of assumptions needed for PFOR to work. Maybe the authors understand this, but even after careful reading - I couldn't really figure it out

---

Many thanks to the authors for their detailed and insightful comments - I have read them and will keep my score as is. I'm still not sure how justified the procedure is without a concrete theoretical result.

---

Update post discussion: After extensive reading and discussions with the other reviewers, I share their concerns about clarity and novelty. In particular, it isnt really clear to me what is being added over the paper by Shalit et al (2017). Based on this, I will lower my score

---

> ### Author Response · Authors · 2022-12-13
> **Response to Post-discussion decision**
>
> Dear Reviewer VvUp,
>
> We noticed your additional concern "It isnt really clear to me what is being added over the paper by Shalit et al (2017)".  We have highlighted the differences and novelties based on Shalit et al (2017) in the article. Examples include, but are not limited to:
>
> `Beginning with counterfactual regression (Shalit et al., 2017) and its revolutionary performance, most prevalent methods handle the selection bias by minimizing the distribution discrepancy between groups in the representation space (see Liuyi et al., 2018; Hassanpour & Greiner, 2020; Cheng et al., 2022). However, two critical issues with these methods have long been neglected, which significantly impedes them from handling the treatment selection bias.`
>
> `Beginning with CFR (Shalit et al., 2017), the unconfoundedness assumption A.1 (see Appendix A) is often taken to circumvent the UCE issue (Ma et al., 2022). However, Assumption A.1 is usually violated in practice as per Figure 3(b), which hinder existing methods including OT from handling treatment selection bias since the backdoor path X′ → T is not blocked.
> `
>
> `
> In particular, CFR-WASS reaches an out-of-sample PEHE of 3.207 on ACIC, significantly outperforming most statistical methods. However, the MSE and UCE issues impede these methods from solving the treatment selection bias. The proposed ESCFR achieves significant improvement over most metrics compared with various prevalent baselines. Combined with the comparisons above, we attribute its superiority to the proposed RMPR and PFOR regularizers, which makes it robust to MSE and UCE. See Appendix D.5 for additional comparison results.
> `
>
> Additionally, a quick and concise response to this question is presented below, which I hope you could kindly read them carefully.
>
> Beginning with Shalit et al (2017) and its revolutionary theoretical framework, representation-based methods seek to learn the representation mapping by minimizing the group discrepancy in the representation space.
>
> Our work, as commented by Reviewer ECPK, "**builds on works in this line, and provided further adjustments to mitigate the mini-batch sampling bias and unobserved confounder problems, which are all well-motivated and significant research questions to investigate in**". Precisely, the works in this line beginning with Shalit et al (2017) have long circumvented **three primary issues that could impede the computation and minimization of the actual group discrepancy and misguide the update of the representation mapping**. We make the first attempt to identify, formulate and mitigate these issues under a generalized optimal transport framework, which is the main contribution and novelty of this study. The following is a summary of the three neglected issues and our corresponding contributions.
>
> (1) Since the majority of works in this line calculate the group discrepancy at a stochastic mini-batch level according to Section 3.1, it is **questionable whether the mini-batch discrepancy is a reasonable and plausible alternative** to the discrepancy calculated with the entire population. To answer this question, Theorem 3.1 demonstrates that PEHE can be optimized by iteratively minimizing the factual outcome estimation error and the mini-batch group discrepancy, supporting the efficacy of the mini-batch group discrepancy.
>
> (2) **Mini-batch sampling effects**, where the works in this line from Shalit et al (2017) fail to quantify the group discrepancy in non-ideal mini-batches, thereby misguiding the update of the representation mapping . RMPR is proposed in Section 3.2 as a solution to this issue.
>
> (3) **Unobserved confounder effects**, where the works in this line neglect the discrepancy of unobserved confounders, and thus highly relies on the strong unconfoundedness assumption. PFOR is proposed in Section 3.3 as a solution to this problem.
>
> Authors.

---

### Official Review · Reviewer_ECPK · 2022-10-24

**Confidence:** 4
**Correctness:** 3
**Technical Novelty And Significance:** 2
**Empirical Novelty And Significance:** 2
**Recommendation:** 5

**Clarity, Quality, Novelty And Reproducibility:**

Overall the paper is written in a relative clear manner, however certain places in writing and the presentation can be improved (see details above). There was some novelty concern in comparison to other representation-based methods in the prior works.
Reproducibility: I was not able to find the implementation code for the submission.

**Strength And Weaknesses:**

Strength:
- Causal effect estimation under treatment selection bias or missing confounders is an important problem, with many downstream applications. Aligning the distribution of each treatment groups in a "learnt" representation space is a natural solution, and has been studied in the prior works. This work builds on works in this line, and provided further adjustments to mitigate the mini-batch sampling bias and unobserved confounder problems, which are all well-motivated and significant research questions to investigate in.
- Using optimal transport appears to be an elegant solution for aligning two distributions. The proposed estimator using optimal transport exhibits simplicity while outperforming several other baselines.
- The proposed estimator was compared with a wide selection of baselines in the prior works

Weakness:
- I think many places in writing for this work can be improved. For example, Figure 2 was actually referenced before Figure 1, and the Figures are in lack of explanations in the caption or text about how the readers should read them. Besides, there were several places in the theoretical analysis where the authors left sentences like "see more rigorous analysis in the appendix" w/t further explanations on what are the exact content to look for, and where to look for in the Appendix. While deferring non-urgent details to the appendix is completely fine, such a way of writing gives the reader a sense that the paper was written maybe in a rush or so.
- It was not very clear what is the key adjustment which avoids the overfitting issue under mini-batches, and the main novelty over (Uri et al 2017) . As the author noted, (Uri et al 2017) minimizes PEHE (eq(3)), which may overfit to the respective group’s properties and thus cannot generalize well to the entire population. However, for the optimal transport solution, suppose that the treated/untreated groups are highly imbalanced, shouldn't that give bias issue to the optimal transport optimization loss too?
- From table 1 the proposed method not only has lower out-of-sample loss, but also the lowest in-sample loss. It was unclear if the other baselines are tuned and trained to the minimal in-sample loss? Otherwise what is the intuition that the proposed estimator also performs the best in the in-sample regime?

**Summary Of The Paper:**

This paper studies the problem of individual causal estimation under treatment selection bias or missing confounders. The authors proposed a new estimator based on representation learning and optimal transport, which aims to minimize certain regularized discrepancy measure in the representation space. The robustness of the proposed estimator under outliers, and the adjusted regularization for missing confounders are analyzed theoretically. The authors further provided empirical study on the performance of the proposed estimator on two datasets.

**Summary Of The Review:**

Overall the paper studies an important and open problem. The proposed solution is intuitive and natural, while the theoretical analysis and experimental comparisons can be improved.



------author rebuttal acknowledgement------
I thank the authors for the detailed response, revising the manuscripts, and answering the questions I raised during my review. I believe this work proposed very reasonable heuristics based on optimal transport and various regularization terms to address the minibatch sampling effect and unobserved confounders problem in causal effects estimation. I also believe that the presentation and empirical evaluations could be further improved to make this work stronger. I keep my original score after the author feedback.

---

> ### Author Response · Authors · 2022-11-19
> **Response to Reviewer ECPK [Follow-up]**
>
> Dear Reviewer ECPK,
>
> Thank you for your sincere and valuable comments again!
>
> Based on your confusions and sincere suggestions, we have completed a substantial editing work to improve the narrative of our figures and highlight the contribution and novelty of this work. Since your engagement and decision are very important to this submission, we would like to follow up to see if our responses and revisions have addressed your main concerns or if you have additional questions. If you have further concerns, we will be more than happy to address them and revise our paper accordingly. Thank you very much!

---

### Official Review · Reviewer_gP25 · 2022-10-24

**Confidence:** 3
**Correctness:** 3
**Technical Novelty And Significance:** 2
**Empirical Novelty And Significance:** 2
**Recommendation:** 6

**Clarity, Quality, Novelty And Reproducibility:**

My evaluation of the quality, clarity and originality of the work is positive although the most novel part in my view: dealing with unobserved confounding, is not sufficiently justified.

**Strength And Weaknesses:**

**Strengths**
- This paper is well-written. All ideas follow intuitively and all quantities are well defined.
- Treatment effect estimation in the presence of unobserved confounding is challenging (if not impossible without strong prior assumptions). I commend the authors for studying it.
- The proposed approach substantially outperforms in the presented experiments.

**Weaknesses**
- Given unobserved confounders, there aren't any assumptions that discuss their influence on the system. For instance, in Sec. 3.3 the authors assume that all variation in outcomes that does not come from observed confounders or treatment, must be due to unobserved confounders. Could it not be due to variables with an independent causal effect on the outcome? I take it that the authors assume the graph in Fig. 2, but still this graph hides the contribution of exogenous variables so that it is not necessarily the case that two units with similar covariates and unobserved confounders have similar outcomes.
- The datasets used in the experiments do not have unobserved confounders (UC), right? How are you evaluating the UC regularization term?


**Summary Of The Paper:**

Treatment effect estimation is a challenging problem that requires generalising to counterfactuals and involves bias and variance due to confounding and lack of overlap between treatment groups. The authors build on the extensive literature studying this problem from the representation learning perspective to propose a new regularisation function with improvements for balancing distributions of treated and untreated populations and dealing with unobserved confounders.

**Summary Of The Review:**

The proposed extensions, in my view, are reasonable heuristics to improve treatment effect estimation although the formalism and theoretical understanding behind them thin (which is understandable given that this problem is challenging). I do believe however that this is a good contribution to the literature.

---

### Official Review · Reviewer_2Gnu · 2022-10-26

**Confidence:** 4
**Correctness:** 2
**Technical Novelty And Significance:** 3
**Empirical Novelty And Significance:** 3
**Recommendation:** 5

**Clarity, Quality, Novelty And Reproducibility:**

See above, I have substantial concerns about the clarity of this paper, which in turn affects the overall quality. However, I do think the paper contains some nice novel ideas.

**Strength And Weaknesses:**

Strengths:
(1) This is an interesting insight regarding minibatches and discrepancy estimation. While I find it unlikely that this effect would render existing methods unusable, the increased variance is certainly an item of concern and the authors do a nice job of addressing it.
(2) The solution is well reasoned and motivated and elegant in execution.
(3) There are strong empirical results.

Weaknesses:
(1) The language in this paper is _very_ difficult to follow. In particular, it is exceedingly loose in some places to the point that it is almost misleading. Some examples:
*  it is always too expensive to conduct randomized experiments. → I think you meant to say often instead of always here?

* I’m not entirely sure what happened but a number of your citations cite either the first name or middle initial instead of last name (e.g. R, et al., Uri).

* “Existing representation-based methods fail to eliminate the treatment selection bias due to the Unobserved Confounder Effects (UCE). “ Is this really what you meant? If so, can you provide specific theoretical evidence of this?

(2) This may partially be an artifact of (1) but the claims in this paper appear to be not terribly well stated and substantiated. This is in contrast with the proposed method which I found to be quite elegant! In particular is the claim that minibatching renders existing discrepancy based learners inconsistent? At which batch sizes? Can we quantify this? The underlying claim that minibatching and the resulting variance is problematic I think is fine, but the authors need to be more careful in the overall language and claims.


**Summary Of The Paper:**

This paper addresses the problem of inferring individual treatment effects from observational data using representation learning via a discrepancy constraint between the representations of the treatment and control groups. In particular, the authors address the problem of dealing with the mini batch sampling which occurs during the deep learning process.

**Summary Of The Review:**

As I stated above, I think this paper overall contains a very interesting idea which serves as a real contribution to the literature. Unfortunately a combination of loose claims and writing make it difficult to engage more fully with the work as it presently stands. To be entirely honest, because of this it wasn't always clear _which_ claims are being made versus are the unfortunate artifact of loose writing. Additionally, I believe that the authors would be well served to clarify exactly what they are correcting for in terms of the _consequences_ of minibatches, not just the presence of minibatching itself. With that being said, I think there is a very nice idea here! I would strongly encourage the authors to make a substantial editing pass to improve the language and narrative of the paper.

---

> ### Author Response · Authors · 2022-11-14
> **Response to Reviewer 2Gnu [1/3]**
>
> We sincerely thank the reviewer for his careful reading and appreciation of our ideas. We have fixed the suggested issues and added missing citations. Here are some responses to specific concerns:
>
> #### **[Comment 1] The language in this paper is very difficult to follow.**
>
> Q1. I strongly encourage the authors to make a substantial editing pass to improve the language and narrative of the paper.
>
> - Thank you very much for your detailed comments on the clarity. We apologize for any unclear presentation and have made a substantial editing pass (1426 total changes, 573 replacements, 344 insertions) to improve the paper's clarity this week.
> - Additionally, we have redesigned most charts and their captions so that the readers can easily get the primary insight into the charts through reading the captions.
> - Making such substantive edits is a long journey, and to this day, a few authors are still working on them. To respond to your comments on time, we have released the most satisfied version. Ongoing detailed revisions will be completed by the end of the discussion phase. We hope these revisions can clarify your confusion and meet with your approval.
>
> Q2. it is `always` too expensive to conduct randomized experiments. → I think you meant to say `often` instead of always here.
>
> - The choice of always here was intended to emphasize the importance of our research problem, i.e., estimating ITE from observational data. We concur that using always would render this article subjective.
> - We have revised the manuscript accordingly. We have also corrected some other expressions that were not objective.
>
> Q3. A number of citations cite the first name or the middle initial instead of the last name.
>
> - Thank you for pointing this out. This oddity is because the authors' names generated by Google Scholar are sometimes inverted. We apologize for not checking this.
> - We have checked all references based on the first authors' home pages. Three incorrect citation formats were found and corrected [1-3].
>
> Q4. "Existing representation-based methods fail to eliminate the treatment selection bias due to the Unobserved Confounder Effects (UCE).
> " Is this really what you meant? If so, can you provide specific theoretical evidence of this?
>
> Yes, that's what we mean, and we can provide two pieces of theoretical evidence.
> - This argument is not first made in this paper. Existing discrepancy-based methods are usually based on the unconfoundedness (conditional exchangeability) assumption, i.e., there is no unobserved confounder effect (UCE). Most of these works emphasize this assumption in the main text, such as assumption 2 in [4], assumption 2.3 in [5], and the "no-hidden confounder assumption" in the introduction of [2]. With this assumption, these methods reasonably circumvent the existence of UCE and limit their scope of application to situations where UCE does not exist.
> - Section 3.3 provides a theoretical discussion of why UCE impedes these methods from handling the treatment selection bias. If their unconfoundedness assumption holds, these discrepancy-based methods could successfully mitigate the treatment selection bias since they block the backdoor path $X\rightarrow T$. However, the violation of this assumption, i.e., the existence of UCE, invalidates these methods since the backdoor path $X^\prime\rightarrow T$ is not blocked, and the treatment selection bias is thus out of control. As a result, existing discrepancy methods fail to eliminate the treatment selection bias due to the UCE.
> - We have revised our manuscript accordingly to highlight these claims.
>
> [1] Judea Pearl, et al. "The book of why: the new science of cause and effect. Basic books." (2018).
>
> [2] Uri Shalit, et al. "Estimating individual treatment effect: generalization bounds and algorithms." ICML (2017).
>
> [3] Soren Reinhold Kunzel, et al. "Metalearners for estimating heterogeneous treatment effects using machine learning." PNAS (2019).
>
> [4] Jing Ma et al. "Learning Causal Effects on Hypergraphs." SIGKDD (2022).
>
> [5] Liuyi Yao, et al. "Representation learning for treatment effect estimation from observational data." NeurIPS (2018).

---

> ### Author Response · Authors · 2022-11-14
> **Response to Reviewer 2Gnu [2/3]**
>
> #### **[Comment 2]  This may partially be an artifact of (1) but the claims in this paper appear to be not well stated and substantiated.**
>
> Q1. Do you claim that minibatching renders existing discrepancy-based learners inconsistent?
>
> Yes, your understanding is correct. Existing discrepancy-based learners can easily fail in tackling the treatment selection bias under mini-batch sampling. We summarize the reasons, from the perspective of both theoretical analysis and empirical analysis, as follows:
> - Theoretically, the sampling complexity term$\mathcal{O}(\cdot)$in Theorem 3.1 indicates the potential risks of bad cases caused by stochastic sampling. Precisely, $\mathcal{O}(\cdot)$ results from the discrepancy between the entire population and the sampled mini-batch units (see Eq.(30) in Appendix A) which is highly dependent on the uncontrollable sampling quality. Therefore, **the discrepancy measure should be robust to bad sampling cases at a mini-batch level; otherwise, the resulting huge variance will impede the computation and reduction of the actual discrepancy.** However, existing methods just ignore this issue.
> - Empirically, Figure 2 shows that Sinkhorn discrepancy can be easily disturbed by many sampling cases, such as outcome imbalance and mini-batch outliers. As a result, the vanilla OT technique **fails to quantify the group discrepancy for producing erroneous transport strategies** in non-ideal mini-batches, and thus **misguides the update of the representation mapping $\psi$**. It makes the prevalent methods, such as the CFR-WASS, suffer greatly from this issue.
> - **Beyond OT**, most prevalent methods based on MMD and adversarial training also suffer from the MSE issue. Given a mini-batch with imbalanced outcomes between treatment groups, for example, these discrepancy-based approaches would naively minimize the discrepancy between groups, erasing discriminative characteristics that are useful for outcome estimation.
> - Notably, an important **advantage** of Sinkhorn discrepancy over other techniques is that it allows us to **formalize the MSE issue** through the mass-preserving constraint which requires that all units in both groups match each other. Additionally, **it provides a grip for handling this issue**: relaxing the marginal constraint and allowing for the creation and destruction of unit's mass, which is the primary intuition of RMPR.
>
> Q2. At which batch sizes? Can we quantify this?
> - In our response to Q1, we have demonstrated that MSE affects the majority of discrepancy-based methods. However, it is difficult to quantify the extent to which MSE affects the performance of these methods, which is actually the **defect of these methods** and beyond the scope of this study.
> - Fortunately, the **matching-based paradigm** of Sinkhorn discrepancy in this study makes it possible to formulate and quantify the MSE issue. We detail how to **quantify the effect of mini-batching with respect to batch size** based on the two cases in Section 3.2.
>   - (a) Outcome imbalance, where units with unrelated factual outcomes would be falsely matched as per Figure 2 (b). The outcome imbalance can be decomposed into two sources: (1) the inherent outcome imbalance between groups in the entire population, which is unrelated to mini-batching (but can also be handled with RMPR); (2) the imbalance caused by mini-batching, which we would discuss in detail here. Consider a case where the outcome is binary and balanced between groups in the entire population. According to **Hoeffding's inequality**, **the larger the batch size the easier it is to obtain an empirical distribution with balanced outcome** between treatment groups. Interestingly, the Hoeffding's inequality is also the core formula for proving Theorem 3.1.
>   - (b) Mini-batch outliers, where outliers of the mini-batch are falsely matched to normal units, causing a substantial disruption of the transportation strategy as per Figure 2 (c).  Its effect on vanilla Sinkhorn discrepancy w.r.t. batch size could be quantified based on the Lemma 1 in [6], where the resulting estimation variance of discrepancy is **lower-bounded by a batchsize-related term**. Additionally, its effect on the Sinkhorn discrepancy with RMPR has been quantified in Theorem 3.2 in our study, which is **upper bounded by a batchsize-related term**.
>
> [6] Kilian Fatras, et al. "Unbalanced minibatch optimal transport; applications to domain adaptation." ICML (2021).

---

> ### Author Response · Authors · 2022-11-14
> **Response to Reviewer 2Gnu [3/3]**
>
> #### Q3. Authors would be well served to clarify exactly what they are correcting for in terms of the consequences of minibatches.
> - Under mini-batch sampling, the existence of bad sampling cases, such as outcome imbalance and outliers, causes huge challenges for existing discrepancy-based methods (see response to Q1 above). Thus, at a mini-batch level, the discrepancy measure should be robust to these bad sampling cases; otherwise, the resulting huge variance will impede it from computing and reducing the actual discrepancy. For example, the vanilla Sinkhorn discrepancy fails to quantify the group discrepancy for producing erroneous transport strategies in non-ideal mini-batches and thus misguides the update of the representation mapping $\psi$. In this work, we aim to correct for the failure of vanilla Sinkhorn discrepancy in non-ideal minibatches.
>
> ---
> #### **Conclusion and Acknowledgement**
>
> Many thanks for expressing your appreciation for our research. We understand that the primary concern is the language and clarity of this paper. We have clarified the main confusion in the response and have made substantial edits accordingly to improve the paper's clarity. The revised version has been uploaded, where important changes are highlighted in blue. We have also provided a detailed comparison with the submitted version in the supporting materials.
>
> Since the revision DDL is approaching, please do let us know if you still have any other concerns that ought to be clarified in revision. Your suggestions would be deeply appreciated!

---

> ### Author Response · Authors · 2022-11-19
> **Response to Reviewer 2Gnu [Follow-up]**
>
> Dear Reviewer 2Gnu,
>
> Thank you for your appreciation for our ideas and methods again!
>
> Following your instructions and sincere comments, we have completed a substantial editing work to improve the narrative of this paper. As your engagement and decision are very important to this submission, we would like to follow up to see if our responses and revisions have addressed your main concerns or if you have additional questions. If our response does not address your concerns, we will be happy to have additional discussions. Thank you very much!

---

### Decision · Program_Chairs · 2023-01-20

**Decision:**

Reject

**Justification For Why Not Higher Score:**

There were serious concerns about the evaluation of the proposal in the paper.

**Justification For Why Not Lower Score:**

N/A

**Metareview: Summary, Strengths And Weaknesses:**

The paper proposed some methods based on optimal transport + different regularization terms to address (1) the mini-batch sampling effect (2) the unobserved confounding problem in estimating causal effects.

Strengths:

The paper works on important problems of unobserved confounding and mini-batch sampling effect.

Weaknesses:

The evaluation of how the proposed method can address the issue of unobserved confounding and mini-batch sampling is lacking. The experiments conducted in the paper did not directly test upon these issues: for example, using the semi-synthetic dataset one could vary the sampling batch sizes or mask one covariate as a hidden confounder to stress test the proposed method.

The paper in its current form is not ready for publication. We encourage the authors to expand the experimental evaluation and submit to a future venue.


**Summary Of Ac-Reviewer Meeting:**

The reviewers were not able to find a time to meet. There is no slot in which more than two reviewers are available. But we were able to have a reviewer discussion over email which all reviewers participated.

Reviews mainly had significant concerns about the evaluations of the paper. The experiments conducted in the paper did not directly test upon the mini-batch issue and the unobserved confounding issue: for example, using the semi-synthetic dataset one could vary the sampling batch sizes or mask one covariate as a hidden confounder to stress test the proposed method. There are also no principled insights on how one should tune over the different regularization strengths as there are many (lambda, epsilon, kappa, gamma as in Eq(15));

There were also some concerns around the baseline comparisons. In section 4.1, the authors stated that "all neural models are trained with the learning rate and weight decay both set to 0.001". However, learning rate and weight decay are hyper-parameters that affect the performance of the models and should be tuned over for each baseline. Given that the in-sample loss for other baselines appears very high, it is unclear if these values are selected in favor of the proposed OT model.

There were also some novelty concerns about which component of the proposal solves the proposed problem. It appears that it is mainly the use of representation that enables the performance gain. However, the use of representation in this context dates back to Shalit et al(2017).